# ssToken: Self-modulated and Semantic-aware Token Selection for LLM Fine-tuning

**Xiaohan Qin**[1,2], **Xiaoxing Wang**[1,*], **Ning Liao**[1], **Cancheng Zhang**[1], **Xiangdong Zhang**[1],
**Mingquan Feng**[1], **Jingzhi Wang**[1], **Junchi Yan**[1,2] [*]
[1]School of AI, Shanghai Jiao Tong University, [2]Shanghai Innovation Institute

## Abstract

Data quality plays a critical role in enhancing supervised fine-tuning (SFT) for large language models (LLMs), and token-level data selection has emerged as a promising direction for its fine-grained nature. Despite their strong empirical performance, existing token-level selection methods share two key limitations: (1) requiring training or accessing an additional reference model, and (2) relying solely on loss information for token selection, which cannot well preserve semantically important tokens that are not favored by loss-based metrics. To address these challenges, we propose **ssToken**, a **S**elf-modulated and **S**emantic-aware **Token** Selection approach. ssToken leverages readily accessible history models to compute the per-token loss difference with the current model, which serves as a self-modulated signal that enables the model to adaptively select tokens along its optimization trajectory, rather than relying on excess loss from an offline-trained reference model as in prior works. We further introduce a semantic-aware, attention-based token importance estimation metric, orthogonal to loss-based selection and providing complementary semantic information for more effective filtering. Extensive experiments across different model families and scales demonstrate that both self-modulated selection and semantic-aware selection alone outperform full-data fine-tuning, while their integration—ssToken—achieves synergistic gains and further surpasses prior token-level selection methods, delivering performance improvements while maintaining training efficiency. Source code is available at https://github.com/jianke0604/ssToken.

## 1 Introduction

Large language models (LLMs) have achieved remarkable progress in recent years (Wu et al., 2025; Xiong et al., 2025; Yang et al., 2026), and supervised fine-tuning (SFT) (Wei et al., 2021; Ouyang et al., 2022; Li et al., 2025b) has emerged as a crucial stage in their modern training pipeline, enhancing instruction-following capability and practical utility. While early efforts primarily focused on scaling up instruction-tuning corpora, sometimes with millions of samples (Longpre et al., 2023; Chung et al., 2024), recent findings have consistently demonstrated that *data quality outweighs data quantity* in SFT (Chen et al., 2023a; Zhou et al., 2024; Fu et al., 2025), as small but carefully curated datasets often yield stronger downstream performance than much larger but noisier collections. Therefore, identifying and prioritizing subsets of data that contribute most to model improvement from massive candidate data pools has become a key challenge in LLM supervised fine-tuning.

Recent studies (Lin et al., 2024; Simoulin et al., 2025; Li et al., 2025a) have shown that even after rigorous sample-level filtering, high-quality datasets still contain substantial token-level noise that degrades training outcomes. Non-task-related patterns or phrases are often redundant or uninformative, and continuing to fine-tune on them offers limited benefit while potentially undermining downstream task performance. This has motivated growing interest in token-level data selection, which aims to refine training datasets at a finer granularity by filtering individual tokens rather than

---

[*]Correspondence author. This work was partly supported by National Natural Science Foundation of China (No. 92370201) and Fundamental and Interdisciplinary Disciplines Breakthrough Plan of the Ministry of Education of China, JYB2025XDXM411.

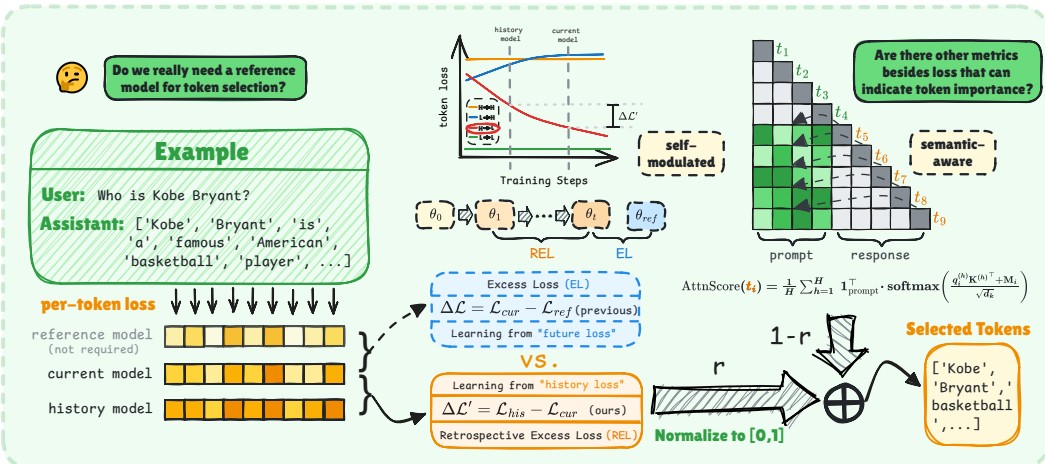

Figure 1: The overall framework of ssToken, which consists of two key components: (i) the self-modulated token selection strategy and (ii) the semantic-aware token importance estimation. We also provide a comparison with prior reference–model–based methods.

entire samples. Importantly, token-level approaches are orthogonal to domain-level and sample-level strategies, thereby broadening their applicability across diverse data selection scenarios.

Despite its advantages, we observe that existing token-level selection methods (Lin et al., 2024; Pang et al., 2025) share two key limitations: (1) requiring training or accessing an additional reference model, and (2) relying solely on loss information for token selection, which cannot well preserve semantically important tokens that are not favored by loss-based metrics. Specifically, recent approaches typically prepare a task-related high-quality dataset and fine-tune either the base model or a smaller proxy model on it to serve as the reference, or alternatively adopt a more powerful model with the same tokenizer guided by prior knowledge. Excess loss is then computed against this reference model to identify task-related tokens that are important for improving downstream performance and instruction-following capability, as well as learnable tokens whose loss steadily decreases. However, directly adopting a stronger model is not always practical, and training an extra reference model incurs additional time and resource costs. Moreover, (Pang et al., 2025) has shown that the performance of the reference model itself has a substantial impact on the effectiveness of token selection. On the other hand, while token-level losses reflect the model's prediction uncertainty, they do not necessarily capture the semantic importance of tokens within their specific context. Frequent yet semantically uninformative tokens may yield similar excess loss values as task-critical ones, making loss-only selection prone to discarding informative content.

In this paper, we propose **ssToken** (**S**elf-modulated and **S**emantic-aware **Token** Selection), a new token-level data selection method designed to address the two limitations discussed above. The core role of a reference model is to identify task-related and learnable tokens while discarding noisy or already well-learned ones. Instead of preparing an additional curated dataset and training a separate reference model, we view the current model itself as a natural teacher: as training progresses, its improvements over its own history provide reliable signals for token selection. As shown in Fig. 1, relative to the history model (e.g., the base model before SFT), the current model has already learned certain task-related patterns and achieved stronger performance, which can be viewed as a reference. This perspective aligns with the principle of self-modulated learning, where the model adaptively regulates tokens that remain informative and learnable along its trajectory. Intuitively, if the current model achieves a notable loss reduction on a token compared with the history model, that token is unlikely to be noisy or already well-learned, but rather represents learnable and informative content. Furthermore, as the current model improves, the history model can be updated adaptively (e.g., via EMA), providing more stable guidance than relying on a fixed reference model.

On the other hand, for a well-pretrained model, the attention matrix over input samples inherently encodes rich semantic information, which we regard as a complementary signal beyond loss for assessing token importance. To leverage this property in the fine-tuning setting, we design an attention-based semantic-aware selection metric that identifies tokens more semantically salient in context and more beneficial for enhancing instruction-following capability. Extensive experiments

show that both self-modulated selection and semantic-aware selection alone outperform full-data fine-tuning, while their integration—ssToken—achieves synergistic gains and further surpasses previous token-level selection methods. Overall, our contributions can be listed as follows:

**1) Self-modulated token selection paradigm.** We replace costly or even unavailable reference models with a self-modulated strategy that leverages the current model's improvements over its history, thereby avoiding additional time and resource costs. Unlike a fixed reference model, the history model can be updated throughout training, providing more stable and long-term guidance.

**2) Semantic-aware token importance estimation metric.** We introduce an attention-based metric that leverages the rich semantic information embedded in attention matrices, offering a complementary signal to loss-based selection. The two signals are orthogonal, and their integration leads to mutually reinforcing effects. Additionally, we design a lightweight implementation that is compatible with efficient attention mechanisms such as FlashAttention, thus introducing negligible computational overhead.

**3) Strong empirical performance across various model families and scales.** We conduct extensive experiments on models ranging from 3B to 14B parameters across multiple benchmarks. Results consistently show that our proposed ssToken not only surpasses full-data fine-tuning baselines by up to **4.3%** but also outperforms previous token-level selection methods by up to **2.8%**, while maintaining training efficiency.

## 2  RELATED WORK

**LLM Data Curation.** The core objective of data curation for LLMs is to maximize training performance and efficiency by enhancing the quality and scale of training data. This encompasses a wide range of strategies, including acquiring data through web crawling (Penedo et al., 2024; Yu et al., 2025) or synthetic generation (Wang et al., 2022; Li et al., 2023; Long et al., 2024; Han et al., 2025; Wang et al., 2025a), filtering (Gao et al., 2020; Penedo et al., 2023) and deduplication (Lee et al., 2021; Kandpal et al., 2022), rebalancing across domains (Xie et al., 2023; Liu et al., 2024), selecting subsets from a given data pool (Liu et al., 2025; Wang et al., 2025b; Fu et al., 2025), and ranking data to enable curriculum learning (Lee et al., 2023; Chen et al., 2023b). In this work, we focus on data selection for LLM supervised fine-tuning.

**LLM Data Selection.** Data selection is a critical component of the LLM data curation pipeline, aiming to further refine filtered and deduplicated high-quality datasets to improve training performance and efficiency (Raffel et al., 2020; Tan et al., 2023; Bukharin & Zhao, 2023; Yu et al., 2024). Sample-level selection has been extensively explored by recent studies (Pang et al., 2024; Li et al., 2024; Fu et al., 2025), and some approaches (Zhang et al., 2024; Wang et al., 2025b) can achieve nearly lossless performance using less than 10% of the original data for fine-tuning. Recently, RHO-1 (Lin et al., 2024) highlights that even carefully curated datasets at the sample level still contain token-level noise and uninformative content. They are the first to explore token-level data selection for LLM training and report substantial gains over full-data baselines. TokenCleaning (Pang et al., 2025) further optimizes token-level selection in the SFT setting. However, these methods all share the two limitations discussed in the introduction. In contrast, our proposed ssToken avoids the need to train or access an extra reference model and leverages attention-based signals to complement loss information with additional semantic cues.

## 3  METHODS

### 3.1  PRELIMINARIES: REFERENCE-MODEL-BASED TOKEN SELECTION

Given a data pool $\mathcal{D}$ with $N$ samples, let $\boldsymbol{x} = \{x_1, x_2, \ldots, x_L\} \in \mathcal{D}$ denote one sample with sequence length $L$. Training a large language model under the next-token prediction paradigm can be formulated as minimizing the negative log-likelihood of observed tokens in the dataset. The loss on a single sample $\boldsymbol{x}$ is defined as

$$\mathcal{L}_\theta(\boldsymbol{x}) = -\frac{1}{|\mathcal{I}|} \sum_{i \in \mathcal{I}} \log \mathbb{P}_\theta(x_i \mid x_{<i}), \tag{1}$$

where $\theta$ denotes the model parameters, $x_i$ is the $i$-th token, and $x_{<i} = \{x_1, \ldots, x_{i-1}\}$ denotes the preceding context. The index set $\mathcal{I} \subseteq \{1, \ldots, L\}$ specifies which tokens are supervised. For pre-training, $\mathcal{I}$ typically includes the entire sequence; for SFT, $\mathcal{I}$ only covers the response tokens, while prompt tokens serve as conditioning context but are not directly supervised.

RHO-1 (Lin et al., 2024) observes that when continuing training on a well-pretrained LLM, token losses can be categorized into four types based on their trajectories: persistent high loss (H→H), increasing loss (L→H), decreasing loss (H→L), and consistently low loss (L→L). It prioritizes task-related tokens that the model can stably learn (H→L), rather than those already mastered (L→L) or those that remain persistently difficult or noisy (H→H and L→H). To this end, a reference model is obtained by further training the base model on a high-quality task-related dataset. Token-level **Excess Loss (EL)** is then computed with respect to this reference model to score and filter tokens. Specifically, the excess loss for token $x_i$ is defined as:

$$\text{EL}(x_i) \;=\; \mathcal{L}_\theta(x_i) - \mathcal{L}_{\theta_{\text{ref}}}(x_i) = \log \frac{\mathbb{P}_{\theta_{\text{ref}}}(x_i \mid x_{<i})}{\mathbb{P}_\theta(x_i \mid x_{<i})}. \tag{2}$$

Here, $\text{EL}(x_i)$ measures the extent to which the prediction probability of token $x_i$ is expected to improve in future training. A larger value indicates that the model is more likely to achieve greater predictive gains on this token. They then rank tokens in the data pool based on EL and their designed scoring strategies, followed by token selection according to a predefined selection ratio. Reference-model-based methods have been empirically shown to deliver substantial improvements over full-data baselines, underscoring the promise of token-level selection while also highlighting the need for more practical and flexible approaches.

## 3.2 SELF-MODULATED TOKEN SELECTION

As discussed in Sec. 1, although reference-model-based token selection demonstrates strong performance, directly adopting a stronger model with the same tokenizer is not always practical, and training a reference model incurs additional time and resource costs. We therefore reinterpret token selection as a *self-modulated process*: rather than depending on an external reference, the model leverages its own training trajectory to progressively determine which tokens remain informative and learnable, thereby aligning token selection with its evolving capability.

Specifically, the history model can be defined as the well-pretrained base LLM before SFT, or updated using checkpoints saved during training. Accordingly, we propose **Retrospective Excess Loss (REL)** as a self-modulated signal that quantifies model progress over its history, serving as a replacement for the EL defined in Sec. 3.1:

$$\text{REL}(x_i) \;=\; \mathcal{L}_{\theta_{\text{his}}}(x_i) - \mathcal{L}_\theta(x_i) = \log \frac{\mathbb{P}_\theta(x_i \mid x_{<i})}{\mathbb{P}_{\theta_{\text{his}}}(x_i \mid x_{<i})}. \tag{3}$$

This formulation operationalizes the self-modulated principle: as the current model improves over its historical states, tokens that continue to yield notable gains are adaptively prioritized, whereas noisy or already well-learned ones are down-weighted. In this way, token selection becomes both reference-free and dynamically aligned with the trajectory of model training. Conceptually, if the current model assigns a substantially higher prediction probability to token $x_i$ than the history model, $x_i$ is more likely to represent content that remains learnable at the current stage, rather than tokens already mastered or inherently noisy.

Moreover, Pang et al. (2025) points out that the capacity of the reference model significantly affects the quality of selected tokens and the final model performance. For training on large-scale datasets, however, providing stable long-term guidance becomes particularly important. Unlike reference models, which are often difficult to update continuously, the history model can be easily iterated through training checkpoints, for example via exponential moving average (EMA):

$$(\text{Optional}) \;\; \theta_{\text{his}}^t = \alpha \, \theta_{\text{his}}^{t-1} + (1 - \alpha) \, \theta^t, \tag{4}$$

where $\alpha$ controls the smoothing factor. In this way, the history model retains information from past parameters while remaining adaptive, thereby enhancing its potential to provide stable guidance for large-scale, long-horizon training.

Although self-modulated token selection eliminates the need to train or access a separate reference model, the history and current models share identical parameters at the beginning of training,

which makes early-stage token selection nearly random. Moreover, similar to prior approaches, self-modulated selection still relies solely on loss information, potentially overlooking tokens that are less favored by loss-based criteria but highly relevant to the task. This limitation motivates us to explore new signals beyond loss that can better capture token importance and task relevance, enabling a more comprehensive token selection strategy.

## 3.3 SEMANTIC-AWARE TOKEN IMPORTANCE ESTIMATION

For a well-pretrained LLM, the attention matrix encodes rich semantic information. When answering task-related questions, a token may exhibit persistently high prediction uncertainty (i.e., high loss), yet its attention signal can still play an essential role in predicting subsequent tokens and thus should not be discarded. Motivated by this, we aim to design an attention-based token importance estimation metric that complements loss-based selection with additional semantic signals. Although prior work has proposed estimating token importance using the sum of attentions from other tokens to the target token (Luo et al., 2020; Adnan et al., 2024), directly applying this approach under the next-token-prediction paradigm with a causal mask introduces additional positional bias, where tokens at different positions receive highly uneven attention. We observe that in the SFT setting, only response tokens need to be selected and used for gradient computation, and all response tokens consistently attend to a fixed-length prompt. This motivates a simple yet effective idea: measuring the total attention each response token assigns to the prompt tokens. Intuitively, since the prompt encodes the task description or instruction to be followed, the extent to which a response token attends to the prompt naturally reflects its task relevance and instruction-following importance, making it a reasonable indicator of token importance.

Specifically, we focus on the attention matrix at a fixed layer $l$ during the forward pass of the model. For each attention head $h \in \{1, \ldots, H\}$ in the decoder, we denote its attention matrix as $A^{(h)} \in \mathbb{R}^{L_{\text{seq}} \times L_{\text{seq}}}$, where $A_{i,j}^{(h)}$ represents the attention weight from token $i$ (query) to token $j$ (key), and $L_{\text{seq}} = L_{\text{prompt}} + L_{\text{resp}}$ is the total sequence length. Let $\mathcal{I}_{\text{prompt}} = \{0, 1, \ldots, L_{\text{prompt}} - 1\}$ and $\mathcal{I}_{\text{resp}} = \{L_{\text{prompt}}, \ldots, L_{\text{seq}} - 1\}$ denote the index sets of prompt and response tokens, respectively. To compute the attentions from response tokens to prompt tokens, we extract the following submatrix:

$$A_{\text{resp} \rightarrow \text{prompt}}^{(h)} := A^{(h)}[\mathcal{I}\text{resp}, \mathcal{I}\text{prompt}] \in \mathbb{R}^{L_{\text{resp}} \times L_{\text{prompt}}}. \tag{5}$$

By summing over the prompt dimension, we obtain the attention scores of all response tokens in the sequence:

$$\text{AttnScore}^{(h)} = A_{\text{resp} \rightarrow \text{prompt}}^{(h)} \cdot \mathbf{1}_{\text{prompt}} \in \mathbb{R}^{L_{\text{resp}}}. \tag{6}$$

Subsequently, we average the attention scores obtained from all heads $h \in \{1, \ldots, H\}$, which serves as a token importance estimator complementary to loss information. In summary, the final attention score for a response token $x_i$ is computed as:

$$\text{AttnScore}(x_i) = \frac{1}{H} \sum_{h=1}^{H} \mathbf{1}_{\text{prompt}}^{\top} \cdot \text{softmax}\left( \frac{q_i^{(h)} \mathbf{K}^{(h)\top} + \mathbf{M}_i}{\sqrt{d_k}} \right), \tag{7}$$

where $\mathbf{M}_i$ denotes the causal mask. An important design question is which layer's attention matrix should be used to compute the attention scores. In Sec. 4.3, we conduct ablation studies comparing early, middle, and deep layers, and conclude that using deeper layers generally yields better results. This finding aligns with prior studies (Aljaafari et al., 2024; Zheng et al., 2024; Rocchetti & Ferrara, 2025), which suggest that semantic representations become increasingly abstract across layers: shallow layers (closer to the input) primarily capture syntax, local structures, and surface-level patterns such as positional relations, bracket matching, and syntactic cues, while deeper layers (closer to the output) focus more on semantic abstraction, high-level concepts, and task-relevant global information, which are typically more influential for instruction following.

Moreover, prior works in other domains that rely on attention scores for analysis (Chen et al., 2024; Ye et al., 2025) are often incompatible with efficient attention implementations such as FlashAttention (Dao et al., 2022). To avoid this, we design a lightweight solution: during the forward pass, we use a hook to store the hidden states of the target layer and then perform a simple recomputation of that layer to retrieve its attention matrix. This design eliminates the need to output full attention matrices during the complete forward pass, thereby making our algorithm fully compatible with efficient attention mechanisms like FlashAttention and ensuring training efficiency.

### 3.4 OVERALL FRAMEWORK OF SSTOKEN

After obtaining both the retrospective excess loss and the attention score for each token during training, we combine them to perform multi-dimensional token selection. Specifically, due to the Transformer's computation mechanism and our definition of attention scores, the attention score is naturally bounded within $[0, 1]$. For REL, we normalize all response tokens within each sample as

$$\text{Normalize}(\text{REL}(x_i)) = \frac{\text{REL}(x_i) - \min_j \text{REL}(x_j)}{\max_j \text{REL}(x_j) - \min_j \text{REL}(x_j)}, \tag{8}$$

which maps the values into the range $[0, 1]$. We then introduce a balance coefficient $\gamma \in [0, 1]$ to weight the two signals and compute the final score for each token:

$$\text{Score}(x_i) = \gamma \cdot \text{Normalize}(\text{REL}(x_i)) + (1 - \gamma) \cdot \text{AttnScore}(x_i). \tag{9}$$

Here, $\gamma$ controls the relative contribution of loss-based and attention-based signals in token selection. When $\gamma = 0$ or $\gamma = 1$, ssToken degenerates into purely attention-based selection or purely loss-based selection, respectively. Our ablation studies in Sec. 4.3 reveal that appropriate weighting consistently outperforms using either signal alone, and the best performance across benchmarks typically emerges when $\gamma$ takes intermediate values rather than the extremes (i.e., 0 or 1). Based on experimental validation, we set $\gamma = 0.5$ as the default choice.

Following (Lin et al., 2024; Pang et al., 2025), we perform token selection with a fixed ratio $\rho$. Specifically, the loss function in Eq. 1 is modified as

$$\mathcal{L}_\theta(\boldsymbol{x}) = -\frac{1}{L_{\text{resp}} \cdot \rho} \sum_i I_\rho(x_i) \log \mathbb{P}_\theta(x_i \mid x_{<i}), \tag{10}$$

where $L_{\text{resp}} \cdot \rho$ denotes the number of response tokens selected from the sample $\boldsymbol{x}$ according to the top-$\rho$ fraction of $\text{Score}(x_i)$. The indicator function $I_\rho(x_i)$ is defined as

$$I_\rho(x_i) = \begin{cases} 1, & \text{if } x_i \text{ is among the top-}\rho \text{ tokens ranked by } \text{Score}(x_i), \\ 0, & \text{otherwise.} \end{cases} \tag{11}$$

Based on prior empirical evidence (Lin et al., 2024; Pang et al., 2025), setting $\rho = 0.6$ is generally effective for retaining task-relevant tokens while filtering out less informative ones. We further provide a detailed ablation study on $\rho$ in Sec. 4.3 and explore an adaptive strategy for adjusting $\rho$ in Sec. 4.4. The complete algorithmic workflow of ssToken is presented in Appendix D.

## 4 EXPERIMENTS

### 4.1 PROTOCOLS

**Data Preparation.** Following the setup in (Pang et al., 2025), we construct our experimental data pool by sampling 50k examples from a collection of five widely used SFT datasets (300k samples in total): Flan_v2 (Longpre et al., 2023), OpenAssistant (Köpf et al., 2024), Stanford Alpaca (Taori et al., 2023), Dolly (Databricks, 2023), and WizardLM (Xu et al., 2023).

**Base Models.** We adopt four widely used open-source LLMs of varying sizes, ranging from 3B to 14B parameters: LLaMA-3.2-3B (Dubey et al., 2024), LLaMA-3.1-8B (Dubey et al., 2024), Qwen-2.5-7B (Hui et al., 2024), and Qwen-2.5-14B (Hui et al., 2024). These models are fine-tuned on our curated data pool and compared against different token selection methods.

**Baselines.** For each model, we evaluate several strategies for comparison: (1) Full-data fine-tuning: training on the entire pool of 50k high-quality samples using supervised fine-tuning. (2) Uniform Random: selecting a fixed proportion of $\rho$ tokens uniformly at random from the data pool for training. (3) RHO-1 (Lin et al., 2024): selecting the top-$\rho$ tokens within each sample based on excess loss. (4) TokenCleaning (Pang et al., 2025): including *Fixed-model Cleaning*, which selects the global top-$\rho$ tokens from the entire data pool according to excess loss, and *Self-evolving Cleaning*, which partitions the data pool and iteratively updates the reference model while selecting global top-$\rho$ tokens from each partition. In our experiments, we reproduce the methods based on their released code and report the best performance between the two variants.

Table 1: Performance comparison of different token selection methods across various models and benchmarks. The best scores are highlighted in **bold**.

| Methods | TriviaQA | TruthfulQA | MMLU | ARC-C | ARC-E | TyDiQA | Wino | HS | LogiQA | AGIEval | AVG |
|---|---|---|---|---|---|---|---|---|---|---|---|
| **Base model: LLaMA-3.2-3B** | | | | | | | | | | | |
| BASE | 50.85 | 39.37 | 56.20 | 42.15 | 74.49 | 19.05 | 69.06 | 55.21 | 29.49 | 30.04 | 46.59 |
| FULL | 52.50 | 43.97 | **57.34** | 45.56 | 76.42 | 39.48 | 70.56 | 55.75 | 29.80 | 32.11 | 50.35 |
| RANDOM | 52.71 | 43.72 | 56.84 | 45.14 | 76.60 | 35.66 | 70.03 | 55.70 | 29.95 | 31.89 | 49.82 |
| RHO-1 | 53.14 | 44.35 | 56.78 | 45.31 | 76.98 | 50.47 | 70.17 | 56.67 | 29.80 | 32.79 | 51.65 |
| TOKENCLEANING | 53.09 | 44.79 | 57.17 | 45.22 | **77.06** | 51.40 | 69.93 | **56.69** | 30.57 | 32.81 | 51.87 |
| SSTOKEN | **54.04** | **47.42** | 57.02 | **45.82** | 76.42 | 51.04 | **70.56** | 56.19 | **32.87** | **33.97** | 52.50 |
| **Base model: LLaMA-3.1-8B** | | | | | | | | | | | |
| BASE | 61.51 | 45.22 | 65.08 | 51.45 | 81.57 | 20.50 | 73.56 | 60.01 | 27.96 | 35.33 | 52.22 |
| FULL | 65.21 | 49.03 | **65.75** | 54.69 | 83.59 | 54.64 | 74.74 | 60.40 | 29.80 | 37.07 | 57.49 |
| RANDOM | 65.23 | 48.75 | 65.68 | 54.27 | 83.38 | 54.13 | 74.59 | 60.32 | 30.41 | 37.41 | 57.42 |
| RHO-1 | 65.39 | 52.80 | 65.50 | **56.23** | **83.88** | 57.83 | 76.48 | 62.46 | 31.80 | 37.70 | 59.01 |
| TOKENCLEANING | 65.83 | 53.41 | 65.28 | 56.06 | 83.84 | 57.13 | **76.87** | **62.55** | 32.64 | 37.81 | 59.14 |
| SSTOKEN | **66.33** | **55.55** | 65.32 | 55.29 | 83.12 | **59.28** | 75.77 | 61.63 | **34.10** | **37.97** | 59.44 |
| **Base model: Qwen-2.5-7B** | | | | | | | | | | | |
| BASE | 50.03 | **56.31** | 74.18 | 48.21 | 80.43 | 26.67 | 73.09 | 59.98 | 30.72 | 56.48 | 55.61 |
| FULL | 56.20 | 50.51 | **74.22** | 50.26 | 81.19 | 63.59 | 73.09 | 59.85 | **31.57** | 56.85 | 59.73 |
| RANDOM | 52.08 | 51.86 | 74.18 | 48.21 | 79.84 | 63.09 | **73.32** | 60.07 | 28.33 | 56.25 | 58.72 |
| RHO-1 | 53.59 | 46.45 | 73.91 | 47.53 | 79.55 | 68.73 | 72.61 | 59.00 | 27.65 | 55.83 | 58.49 |
| TOKENCLEANING | 55.57 | 46.14 | 73.88 | 48.72 | 80.47 | 68.17 | 73.24 | 58.55 | 27.34 | 56.17 | 58.83 |
| SSTOKEN | **56.73** | 50.75 | 74.18 | **50.51** | **81.48** | **69.74** | 72.93 | **60.30** | 31.03 | **57.18** | 60.48 |
| **Base model: Qwen-2.5-14B** | | | | | | | | | | | |
| BASE | 59.42 | **58.45** | 79.80 | 55.80 | 82.32 | 28.89 | 75.06 | 63.36 | 38.10 | 64.27 | 60.55 |
| FULL | 66.36 | 53.77 | 79.75 | 56.40 | 82.83 | 64.59 | 74.98 | **63.37** | 36.87 | 66.06 | 64.50 |
| RANDOM | 63.91 | 53.86 | 79.92 | 56.31 | 82.87 | 65.32 | 75.53 | 63.30 | 39.02 | 66.19 | 64.62 |
| RHO-1 | 65.28 | 52.14 | 79.62 | **56.57** | **83.71** | 71.21 | 75.85 | 63.03 | 35.33 | 65.23 | 64.80 |
| TOKENCLEANING | 66.42 | 51.87 | 79.66 | 55.97 | 83.08 | 69.37 | **76.56** | 62.33 | 38.56 | 64.86 | 64.87 |
| SSTOKEN | **66.55** | 52.96 | **79.97** | 56.22 | 82.49 | **71.94** | 76.24 | 62.98 | **42.70** | **66.30** | 65.84 |

Consistent with the setup in (Pang et al., 2025), for methods requiring a reference model, the reference is trained on a 10k high-quality subset of the data pool, which is obtained through further filtering with the new powerful sample-level selection method $DS^2$ (Pang et al., 2024). The token selection ratio $\rho$ is predetermined, and detailed settings as well as related ablation studies are provided in Sec. 4.2 and Sec. 4.3.

**Evaluation.** We evaluate our method on a diverse suite of benchmarks that cover different aspects of LLM capabilities, including MMLU (Hendrycks et al., 2021), TriviaQA (Joshi et al., 2017), TruthfulQA (Lin et al., 2021), ARC-Easy (Clark et al., 2018), ARC-Challenge (Clark et al., 2018), TyDiQA (Clark et al., 2020), Winogrande (Sakaguchi et al., 2021), HellaSwag (Zellers et al., 2019), LogiQA (Liu et al., 2020), and AGIEval (Zhong et al., 2023). Together, these benchmarks provide a comprehensive assessment of factual knowledge, reasoning ability, and cross-lingual generalization. Evaluations are conducted using the standard `lm-eval-harness`[1] (Gao et al., 2024) framework, and additional training and evaluation details can be found in Appendix B.

## 4.2 MAIN RESULTS

We conduct experiments on four different base models with five token selection strategies (including our proposed ssToken) and evaluate their performance across ten general-domain benchmarks. The detailed results are presented in Table 1. The balance coefficient $\gamma$ is set to 0.5 across all model sizes. The token selection ratio $\rho$ is set to 0.6 for LLaMA-3.2-3B, LLaMA-3.1-8B, and Qwen-2.5-7B, and 0.8 for Qwen-2.5-14B. For the same base model, all token selection methods are evaluated under the same ratio. Detailed ablation studies can be found in Sec. 4.3.

As shown in Table 1, ssToken achieves the best average performance across all four base models compared to other existing token selection methods. Relative to the full-data fine-tuning baseline, ssToken improves the average performance by **4.3%**, **3.4%**, **1.3%**, and **2.1%** on LLaMA-3.2-3B, LLaMA-3.1-8B, Qwen-2.5-7B, and Qwen-2.5-14B, respectively. We also observe that although prior token selection methods RHO-1 and TokenCleaning show strong performance on LLaMA-3.2-3B and LLaMA-3.1-8B, their performance on the Qwen family is only marginally better than—or

---

[1] https://github.com/EleutherAI/lm-evaluation-harness

sometimes even worse than—that of full-data fine-tuning. In contrast, our proposed ssToken consistently delivers strong and stable gains across different model sizes and families, further demonstrating its effectiveness and generality.

Furthermore, we find that the impact of token selection varies across different benchmarks. For knowledge-intensive tasks such as MMLU, ARC-Challenge, and ARC-Easy, which heavily rely on pretraining, token selection during SFT does not lead to notable performance improvements. In contrast, for benchmarks like TyDiQA, TriviaQA, and AGIEval that target downstream tasks or require stronger instruction-following ability, SFT typically yields substantial gains for the base models, and token selection often provides further improvements. Notably, ssToken outperforms previous token selection methods on nearly all QA tasks across the four models. Since QA tasks generally demand stronger instruction-following capability, we attribute this advantage to the attention-based component of ssToken, which complements loss information with rich semantic cues and effectively guides the selection of instruction-relevant tokens.

**Performance vs. Training Time.** In Fig. 2, we further compare the performance and total training time (including model loading and reference model training) of different token selection methods against the full-data fine-tuning baseline across various model sizes. It is worth noting that existing token selection approaches are primarily designed to enhance performance rather than reduce training cost. Specifically, while the filtered-out tokens are masked during loss computation, they still participate in the forward pass, and thus these methods do not yield reductions in training time compared to full-data fine-tuning. As shown in the figure, while RHO-1 and TokenCleaning achieve notable performance gains over the full-data baseline, their reliance on a reference model incurs substantial additional time and resource overhead. In contrast, ssToken does not need to train an extra reference model, and its lightweight

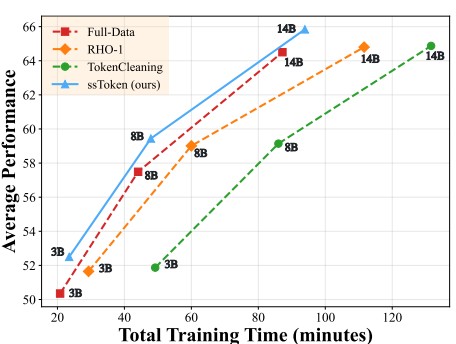

Figure 2: Average performance vs. total training time across different methods.

design for obtaining attention matrices introduces marginal computational cost. Consequently, compared with the full-data baseline, ssToken delivers significant performance improvements with only a marginal increase in training time.

## 4.3 ABLATION STUDIES

Our proposed ssToken involves a key hyperparameter, the balance coefficient $\gamma$, which controls the relative contributions of loss-based and attention-based information. In addition, similar to previous token selection methods, ssToken also requires a predefined ratio $\rho$ for selecting tokens. We therefore conduct experiments on these two hyperparameters across models of different sizes, aiming to investigate their general effects on model performance and to provide practical guidance for their appropriate settings.

**Ablation studies on the balance coefficient $\gamma$.** As shown in Fig. 3a, compared with the full-data fine-tuning baseline, both self-modulated token selection ($\gamma = 1$) and semantic-aware token selection ($\gamma = 0$) achieve superior performance, while combining the two signals in proper proportions yields synergistic effects and further improvements. This validates our hypothesis that loss-based and attention-based information capture complementary features: the latter provides additional semantic cues beyond what is reflected in loss values.

From the results, we observe that $\gamma = 0.5$ achieves the best performance for models with 8B and 14B parameters, and ranks second for the 3B model (LLaMA-3.2-3B), slightly behind $\gamma = 0.75$. A closer look at the detailed results in Appendix C.1 shows that the advantage of $\gamma = 0.75$ on LLaMA-3.2-3B mainly comes from its strong performance on TyDiQA and TruthfulQA. In contrast, although $\gamma = 0.5$ is marginally weaker than $\gamma = 0.75$ on LLaMA-3.2-3B, it achieves more balanced performance across benchmarks. Considering these observations, together with its optimal results on larger models (8B and 14B), we recommend setting $\gamma = 0.5$ as the default choice.

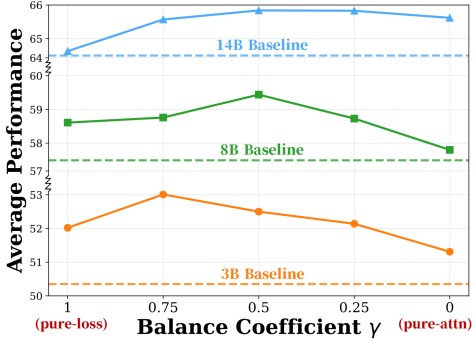 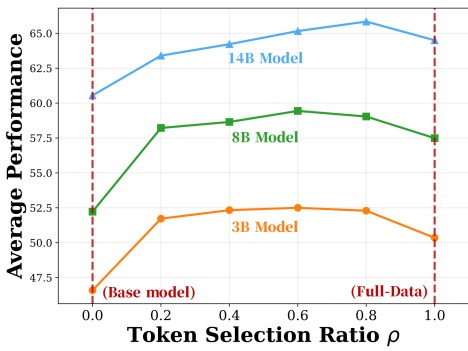

(a) Ablation study on the balance coefficient $\gamma$         (b) Ablation study on token selection ratio $\rho$

Figure 3: Ablation studies on $\gamma$ and $\rho$. (a) Varying the balance coefficient $\gamma$, with dashed lines indicating the full-data fine-tuning baselines for different model sizes. (b) Varying the token selection ratio $\rho$. Both plots report average performance across model sizes (3B, 8B, 14B).

**Ablation studies on the token selection ratio $\rho$.** As shown in the Fig. 3b, although the SFT data we use has already been curated through strong data cleaning pipelines, applying token-level selection with an appropriate ratio still yields further improvements over full-data fine-tuning across different model sizes. For LLaMA-3.2-3B and LLaMA-3.1-8B, the best results are achieved at $\rho = 0.6$, consistent with the ablation findings reported in (Pang et al., 2025). However, for Qwen-2.5-14B, we observe that not only ssToken but also other token selection methods (Random, RHO-1, Token-Cleaning) reach their peak performance at $\rho = 0.8$ (detailed results on this comparison are presented in Appendix C.1). We hypothesize that this may stem from differences in pretraining data distributions, which affect the capability boundaries of different model families; in addition, larger models may better capture more challenging task-related patterns that smaller models fail to recognize. In summary, the choice of $\rho$ should be guided by both model capacity and data quality. In line with prior works (Lin et al., 2024; Pang et al., 2025), our ablations suggest that $\rho = 0.6$ is generally a robust choice, while slightly larger values may yield further improvements in certain settings.

**Ablation Study on Attention Layer Depth.** In our semantic-aware token importance estimation, we require the attention matrix from a specific layer of the forward pass, and the choice of layer can influence the effectiveness of token selection. To investigate this, we conduct experiments on LLaMA-3.2-3B and Qwen-2.5-7B by extracting attention matrices from shallow, medium, and deep layers, respectively. The results are reported in Table 2, with the balancing coefficient $\gamma$ fixed at 0.5 and the token selection ratio $\rho$ set to 0.6.

Across both model families, several consistent patterns emerge. First, for knowledge-intensive benchmarks that rely on pretraining (e.g., MMLU), different attention layers do not exhibit a strictly dominant winner. In contrast, for QA-style tasks (e.g., TruthfulQA, TyDiQA) and logical/common-sense reasoning benchmarks (e.g., LogiQA, HellaSwag), deep-layer attention consistently performs best. More concretely, on LLaMA-3.2-3B, deep-layer attention achieves the best score on **7** out of **10** tasks, and on Qwen-2.5-7B, it achieves the best score on **6** out of **10** tasks. In both models, the deep-layer variant also achieves the highest overall average score.

As discussed in Sec. 3.3, this pattern aligns with observations from prior interpretability and probing studies (Aljaafari et al., 2024; Zheng et al., 2024): shallow layers primarily encode local syntactic or positional features, whereas deeper layers capture more abstract semantic structure and task-relevant dependencies. Overall, these results demonstrate that using deep-layer attention provides a stable and semantically meaningful signal for token selection, and performs well across model families and task types without the need for task-specific tuning. We therefore adopt deep-layer attention as the default configuration in ssToken.

### 4.4 EXPLORING ADAPTIVE TOKEN SELECTION RATIO

In line with prior token-level selection methods, ssToken uses a predefined token selection ratio $\rho$ as the threshold for filtering informative tokens. While Sec. 4.3 presents a comprehensive study of different fixed values of $\rho$, the optimal setting may vary across model families, training stages, or

Table 2: Ablation study on attention layer depth on LLaMA-3.2-3B and Qwen-2.5-7B. Results are reported on ten benchmarks, with the best score among ssToken variants highlighted in **bold**.

| Methods | TriviaQA | TruthfulQA | MMLU | ARC-C | ARC-E | TyDiQA | Wino | HS | LogiQA | AGIEval | AVG |
|---|---|---|---|---|---|---|---|---|---|---|---|
| *Base model: LLaMA-3.2-3B (28 layers)* | | | | | | | | | | | |
| BASE | 50.85 | 39.37 | 56.20 | 42.15 | 74.49 | 19.05 | 69.06 | 55.21 | 29.49 | 30.04 | 46.59 |
| FULL-DATA | 52.50 | 43.97 | 57.34 | 45.56 | 76.42 | 39.48 | 70.56 | 55.75 | 29.80 | 32.11 | 50.35 |
| SHALLOW (LAYER 1) | 54.36 | 45.12 | 56.23 | 45.48 | **77.48** | 48.52 | 70.09 | 53.76 | 31.80 | 33.43 | 51.63 |
| MEDIUM (LAYER 14) | **54.45** | 47.04 | **57.43** | 45.73 | 77.23 | 49.20 | 69.77 | 55.40 | 29.95 | 32.96 | 51.92 |
| DEEP (LAYER 28) | 54.04 | **47.42** | 57.02 | **45.82** | 76.42 | **51.04** | **70.17** | **56.19** | **32.87** | **33.97** | **52.50** |
| *Base model: Qwen-2.5-7B (28 layers)* | | | | | | | | | | | |
| BASE | 50.03 | 56.31 | 74.18 | 48.21 | 80.43 | 26.67 | 73.09 | 59.98 | 30.72 | 56.48 | 55.61 |
| FULL-DATA | 56.20 | 50.51 | 74.22 | 50.26 | 81.19 | 63.59 | 73.09 | 59.85 | 31.57 | 56.85 | 59.73 |
| SHALLOW (LAYER 1) | 55.47 | 46.25 | **74.26** | 49.23 | 81.10 | 68.70 | 72.69 | 59.32 | 27.90 | 57.06 | 59.20 |
| MEDIUM (LAYER 14) | **57.20** | 48.40 | 73.78 | **51.47** | **82.58** | 68.94 | 72.74 | 58.31 | 29.62 | 55.26 | 59.83 |
| DEEP (LAYER 28) | 56.73 | **50.75** | 74.18 | 50.51 | 81.48 | **69.74** | **72.93** | **60.30** | **31.03** | **57.18** | **60.48** |

sample quality. This motivates us to explore whether adaptively adjusting $\rho$ during training can offer further benefits beyond a manually chosen constant. To this end, we introduce a simple training-step–based decay schedule:

$$\rho_t = \rho_{\min} + (\rho_{\max} - \rho_{\min}) \left(1 - \frac{t}{T}\right)^{\beta},$$

where $\rho_{\max}$ and $\rho_{\min}$ are fixed hyperparameters but offer greater flexibility in setting, as they do not require rigorous ablation experiments to determine like the fixed selection ratio. The rationale behind this design is intuitive. At the early stage of training, model predictions are still unstable, and using a relatively large $\rho$ (e.g., $\approx 0.8$) avoids prematurely discarding potentially useful tokens when the semantic and loss-based signals have not yet fully matured. As training progresses and the model becomes increasingly capable of identifying informative patterns, a gradually decreasing $\rho$ encourages the model to concentrate more on high-value tokens. In the late stage, a smaller $\rho$ (e.g., $\approx 0.4$–$0.5$) sharpens this focus and helps the model refine the most semantically and instruction-relevant components of the data. This decay process naturally creates an online curriculum in which the effective training subset shrinks as model capacity improves.

We evaluate this adaptive schedule on LLaMA-3.2-3B (with $\beta = 1$), and the detailed results are presented in Table 6 in the Appendix. The adaptive strategy consistently outperforms all predefined fixed values of $\rho$ considered in our ablations (i.e., $\rho \in 0, 0.2, 0.4, 0.6, 0.8, 1.0$), highlighting the potential of dynamically adjusting $\rho$ and pointing to adaptive token selection as a promising direction for future work.

## 5 CONCLUSION, LIMITATIONS AND FUTURE WORK

In this paper, we introduced **ssToken**, a self-modulated and semantic-aware token selection framework for LLM supervised fine-tuning. By formulating token selection as a self-modulated process, ssToken leverages retrospective excess loss over historical model states to adaptively prioritize tokens that remain informative and learnable, thereby eliminating the need for training an additional reference model. Moreover, by leveraging an attention-based score tailored for the SFT setting, ssToken provides a semantic complement to loss-based information. Experiments demonstrate that both self-modulated selection and semantic-aware selection alone outperform full-data fine-tuning, while combining the two signals yields synergistic gains and further improvements. Extensive results across different model families and scales show that ssToken consistently surpasses both full-data baselines and prior token selection methods, achieving performance improvements while maintaining training efficiency.

**Limitations and Future Work.** Similar to previous token selection methods, ssToken requires a predefined token selection ratio $\rho$ to filter informative tokens. In Sec. 4.4, we explore a simple training-stage–based adaptive schedule for $\rho$ and empirically verify its potential. Beyond adjusting $\rho$ according to training progress, another promising direction is to make $\rho$ dependent on model behavior or data characteristics—such as token-level loss dynamics or sample quality—thereby enabling truly data-driven and model-aware token selection. We leave the development of such more flexible adaptive mechanisms for future work.

## ETHICS STATEMENT

We are committed to conducting this research in line with responsible and ethical AI practices. All experiments are performed on publicly available datasets that are widely used in the community, and no private or sensitive user data is involved. The proposed method, ssToken, aims to improve the efficiency and performance of large language model fine-tuning without introducing additional risks beyond those already associated with LLMs. Nevertheless, we acknowledge the dual-use nature of language technologies and encourage responsible application of our method in accordance with established community guidelines for safe and ethical AI development.

## REPRODUCIBILITY STATEMENT

All base models, datasets, and evaluation tools used in this work are publicly available and widely adopted in the community. As detailed in Sec. 4, we provide clear descriptions of the models, data sources, and evaluation benchmarks employed. Comprehensive training details, including learning rate schedules, batch sizes, and optimization settings, are further specified in Appendix B. In addition, all hyperparameters analyzed in the paper have been thoroughly examined through ablation studies to ensure the robustness of our conclusions. To further facilitate reproducibility, we have anonymously released the complete source code at https://github.com/jianke0604/ssToken.

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

## A  THE USE OF LARGE LANGUAGE MODELS

During the preparation of this manuscript, we employed large language models solely to aid or polish the writing. The core ideas, algorithmic design, code implementation, experiments, and overall manuscript writing were entirely carried out by the authors.

## B  EXPERIMENT DETAILS

### B.1  DATA POOL

Following (Pang et al., 2025), we construct our data pool from five commonly adopted SFT datasets, which are either manually curated or generated with the assistance of powerful LLMs. These datasets vary significantly in terms of prompt style, annotation quality, average length, and task focus, thereby ensuring that the resulting pool is both representative and diverse. To maintain consistency across sources, we standardize all datasets using the template introduced by Wang et al. (2023), which explicitly marks conversational roles with `<|User|>` and `<|Assistant|>` tags.

### B.2  EVALUATION BENCHMARKS

We evaluate models on a wide range of benchmarks that capture different aspects of LLM capability, again following (Pang et al., 2025). For datasets such as MMLU, LogiQA, ARC-C, ARC-E, Winogrande, AGIEval and HellaSwag, we report accuracy as the evaluation metric. In particular, for TruthfulQA, which is formulated as a multiple-choice task, we adopt the MC2 metric that scores only the answer assigned the highest probability by the model. For TriviaQA, we use the *exact_match* metric. For TyDiQA, we use the 1-shot F1 score. All evaluations are performed using the official `lm-evaluation-harness` toolkit, with the default setting of evaluating on the full benchmark sample set.

### B.3  TRAINING DETAILS

Following the setup in (Pang et al., 2025), we adopt the LoRA technique (Hu et al., 2022) with a rank of 64 and a scaling factor of 16. Training is conducted for one epoch with a total batch size of 48 and a learning rate of $1 \times 10^{-4}$. The maximum input sequence length is set to 2048 tokens. All experiments are conducted on eight NVIDIA H200 GPUs. For the training time reported in Fig. 2, we present the complete runtime of each token selection method, including both model loading and the training of the reference model. Due to the relatively small size of the SFT data pool, we observe that enabling the optional EMA-based updating of the history model does not yield noticeable performance gains, while model merging introduces additional computational overhead. Therefore, in all experiments we use a fixed base model as the history model. We hypothesize that adaptive updating of the history model may demonstrate greater potential in large-horizon training scenarios.

For computing REL in Eq. 3, we do not load the history and current models simultaneously nor perform two forward passes. Instead, we adopt a lightweight two-stage implementation: we first run the history model once over the training corpus to cache token-level losses, and during SFT the current model simply reads these cached values and computes REL via a single tensor subtraction. This preprocessing step requires only a few minutes and avoids any increase in peak GPU memory, making ssToken incur only marginal overhead compared with full-data fine-tuning.

## C  ADDITIONAL EXPERIMENTAL RESULTS

### C.1  DETAILED ABLATION ON $\gamma$ AND $\rho$

In Sec. 4.3 of the main text, we presented an ablation study on the hyperparameters $\gamma$ and $\rho$, with the final results briefly summarized in Fig. 3. In this section, we provide the complete experimental results of ssToken with respect to $\gamma$ and $\rho$, reported in Table. 3 and Table. 4. In addition, Table. 5 compares the performance of different token selection methods on Qwen-2.5-14B under two settings, $\rho = 0.6$ and $\rho = 0.8$.

Table 3: Performance of SSTOKEN with different balance coefficient $\gamma$ across different base models and various benchmarks.

| Methods | TriviaQA | TruthfulQA | MMLU | ARC-C | ARC-E | TyDiQA | Wino | HS | LogiQA | AGIEval | AVG |
|---|---|---|---|---|---|---|---|---|---|---|---|
| Base model: LLaMA-3.2-3B | | | | | | | | | | | |
| BASE | 50.85 | 39.37 | 56.20 | 42.15 | 74.49 | 19.05 | 69.06 | 55.21 | 29.49 | 30.04 | 46.59 |
| FULL | 52.50 | 43.97 | 57.34 | 45.56 | 76.42 | 39.48 | 70.56 | 55.75 | 29.80 | 32.11 | 50.35 |
| SSTOKEN ($\gamma = 1.0$) | 51.04 | 48.34 | 56.34 | 43.26 | 74.66 | 56.97 | 68.51 | 51.78 | 34.87 | 34.39 | 52.02 |
| SSTOKEN ($\gamma = 0.75$) | 52.64 | 51.86 | 56.72 | 44.97 | 75.97 | 55.64 | 70.17 | 54.50 | 33.03 | 34.60 | 53.01 |
| SSTOKEN ($\gamma = 0.5$) | 54.04 | 47.42 | 57.02 | 45.82 | 76.42 | 51.04 | 70.17 | 56.19 | 32.87 | 33.97 | 52.50 |
| SSTOKEN ($\gamma = 0.25$) | 54.46 | 46.62 | 57.31 | 45.82 | 77.36 | 49.11 | 70.72 | 55.75 | 31.34 | 32.87 | 52.14 |
| SSTOKEN ($\gamma = 0.0$) | 53.32 | 45.35 | 56.93 | 45.48 | 77.10 | 47.27 | 70.32 | 55.75 | 29.65 | 31.93 | 51.31 |
| Base model: LLaMA-3.1-8B | | | | | | | | | | | |
| BASE | 61.51 | 45.22 | 65.08 | 51.45 | 81.57 | 20.50 | 73.56 | 60.01 | 27.96 | 35.33 | 52.22 |
| FULL | 65.21 | 49.03 | 65.75 | 54.69 | 83.59 | 54.64 | 74.74 | 60.40 | 29.80 | 37.07 | 57.49 |
| SSTOKEN ($\gamma = 1.0$) | 65.50 | 56.51 | 64.70 | 53.33 | 82.07 | 60.60 | 74.90 | 56.78 | 32.87 | 38.81 | 58.61 |
| SSTOKEN ($\gamma = 0.75$) | 65.87 | 54.24 | 65.63 | 53.75 | 82.07 | 59.23 | 75.69 | 59.93 | 32.87 | 38.28 | 58.76 |
| SSTOKEN ($\gamma = 0.5$) | 66.33 | 55.55 | 65.32 | 55.29 | 83.12 | 59.28 | 75.77 | 61.63 | 34.10 | 37.97 | 59.44 |
| SSTOKEN ($\gamma = 0.25$) | 65.54 | 53.57 | 65.63 | 55.29 | 83.25 | 58.40 | 75.45 | 60.24 | 32.72 | 37.19 | 58.73 |
| SSTOKEN ($\gamma = 0.0$) | 65.72 | 48.87 | 65.83 | 54.86 | 83.63 | 54.67 | 75.22 | 60.60 | 30.88 | 37.74 | 57.80 |
| Base model: Qwen-2.5-14B | | | | | | | | | | | |
| BASE | 59.42 | 58.45 | 79.80 | 55.80 | 82.32 | 28.89 | 75.06 | 63.36 | 38.10 | 64.27 | 60.55 |
| FULL | 66.36 | 53.77 | 79.75 | 56.40 | 82.83 | 64.59 | 74.98 | 63.37 | 36.87 | 66.06 | 64.50 |
| SSTOKEN ($\gamma = 1.0$) | 64.41 | 52.62 | 79.69 | 55.38 | 82.45 | 69.31 | 75.61 | 63.01 | 37.64 | 66.18 | 64.63 |
| SSTOKEN ($\gamma = 0.75$) | 66.64 | 52.15 | 77.93 | 56.40 | 83.21 | 70.91 | 75.69 | 63.10 | 43.32 | 66.30 | 65.57 |
| SSTOKEN ($\gamma = 0.5$) | 66.55 | 52.96 | 79.97 | 56.22 | 82.49 | 71.94 | 76.24 | 62.98 | 42.70 | 66.30 | 65.84 |
| SSTOKEN ($\gamma = 0.25$) | 67.04 | 53.17 | 79.90 | 55.55 | 82.32 | 72.30 | 75.85 | 62.91 | 43.16 | 66.10 | 65.83 |
| SSTOKEN ($\gamma = 0.0$) | 66.16 | 54.06 | 79.92 | 56.14 | 83.12 | 73.53 | 76.01 | 63.22 | 37.94 | 66.11 | 65.62 |

Table 4: Performance of ssToken under different token selection ratios $\rho$ across different base models and various benchmarks.

| Methods | TriviaQA | TruthfulQA | MMLU | ARC-C | ARC-E | TyDiQA | Wino | HS | LogiQA | AGIEval | AVG |
|---|---|---|---|---|---|---|---|---|---|---|---|
| Base model: LLaMA-3.2-3B | | | | | | | | | | | |
| BASE ($\rho = 0$) | 50.85 | 39.37 | 56.20 | 42.15 | 74.49 | 19.05 | 69.06 | 55.21 | 29.49 | 30.04 | 46.59 |
| FULL ($\rho = 1$) | 52.50 | 43.97 | 57.34 | 45.56 | 76.42 | 39.48 | 70.56 | 55.75 | 29.80 | 32.11 | 50.35 |
| SSTOKEN ($\rho = 0.2$) | 51.70 | 53.82 | 55.63 | 44.20 | 73.65 | 47.50 | 70.48 | 54.32 | 33.03 | 32.91 | 51.72 |
| SSTOKEN ($\rho = 0.4$) | 51.79 | 51.62 | 56.22 | 45.65 | 75.55 | 50.70 | 70.80 | 55.73 | 31.80 | 33.47 | 52.33 |
| SSTOKEN ($\rho = 0.6$) | 54.04 | 47.42 | 57.02 | 45.82 | 76.42 | 51.04 | 70.17 | 56.19 | 32.87 | 33.97 | 52.50 |
| SSTOKEN ($\rho = 0.8$) | 53.99 | 45.72 | 57.49 | 45.65 | 77.57 | 50.10 | 70.24 | 56.26 | 31.95 | 33.88 | 52.29 |
| Base model: LLaMA-3.1-8B | | | | | | | | | | | |
| BASE ($\rho = 0$) | 61.51 | 45.22 | 65.08 | 51.45 | 81.57 | 20.50 | 73.56 | 60.01 | 27.96 | 35.33 | 52.22 |
| FULL ($\rho = 1$) | 65.21 | 49.03 | 65.75 | 54.69 | 83.59 | 54.64 | 74.74 | 60.40 | 29.80 | 37.07 | 57.49 |
| SSTOKEN ($\rho = 0.2$) | 66.60 | 57.28 | 65.62 | 52.47 | 80.93 | 56.45 | 74.98 | 56.48 | 32.41 | 38.95 | 58.22 |
| SSTOKEN ($\rho = 0.4$) | 66.70 | 55.42 | 64.98 | 53.41 | 82.15 | 57.98 | 75.14 | 57.38 | 36.25 | 37.11 | 58.65 |
| SSTOKEN ($\rho = 0.6$) | 66.33 | 55.55 | 65.32 | 55.29 | 83.12 | 59.28 | 75.77 | 61.63 | 34.10 | 37.97 | 59.44 |
| SSTOKEN ($\rho = 0.8$) | 65.60 | 52.13 | 65.22 | 54.95 | 83.38 | 61.69 | 75.22 | 61.77 | 33.18 | 37.25 | 59.04 |
| Base model: Qwen-2.5-14B | | | | | | | | | | | |
| BASE ($\rho = 0$) | 59.42 | 58.45 | 79.80 | 55.80 | 82.32 | 28.89 | 75.06 | 63.36 | 38.10 | 64.27 | 60.55 |
| FULL ($\rho = 1$) | 66.36 | 53.77 | 79.75 | 56.40 | 82.83 | 64.59 | 74.98 | 63.37 | 36.87 | 66.06 | 64.50 |
| SSTOKEN ($\rho = 0.2$) | 68.71 | 53.88 | 78.57 | 56.66 | 82.49 | 61.34 | 75.61 | 59.41 | 33.03 | 64.33 | 63.40 |
| SSTOKEN ($\rho = 0.4$) | 64.31 | 52.71 | 79.68 | 54.61 | 81.73 | 70.33 | 75.61 | 61.76 | 36.41 | 65.05 | 64.22 |
| SSTOKEN ($\rho = 0.6$) | 64.07 | 52.90 | 79.88 | 56.57 | 83.63 | 71.94 | 75.85 | 62.62 | 38.71 | 65.46 | 65.16 |
| SSTOKEN ($\rho = 0.8$) | 66.55 | 52.96 | 79.97 | 56.22 | 82.49 | 71.94 | 76.24 | 62.98 | 42.70 | 66.30 | 65.84 |

Table 5: Comparison of token selection methods on Qwen-2.5-14B at two ratios, $\rho = 0.6$ and $\rho = 0.8$.

| Methods | TriviaQA | TruthfulQA | MMLU | ARC-C | ARC-E | TyDiQA | Wino | HS | LogiQA | AGIEval | AVG |
|---|---|---|---|---|---|---|---|---|---|---|---|
| Base model: Qwen-2.5-14B | | | | | | | | | | | |
| BASE | 59.42 | 58.45 | 79.80 | 55.80 | 82.32 | 28.89 | 75.06 | 63.36 | 38.10 | 64.27 | 60.55 |
| FULL | 66.36 | 53.77 | 79.75 | 56.40 | 82.83 | 64.59 | 74.98 | 63.37 | 36.87 | 66.06 | 64.50 |
| RANDOM ($\rho = 0.6$) | 65.91 | 53.57 | 79.85 | 56.31 | 83.50 | 63.32 | 75.22 | 63.33 | 33.33 | 66.10 | 64.04 |
| RANDOM ($\rho = 0.8$) | 63.91 | 53.86 | 79.92 | 56.31 | 82.87 | 65.32 | 75.53 | 63.30 | 39.02 | 66.19 | 64.62 |
| RHO-1 ($\rho = 0.6$) | 62.16 | 51.71 | 79.55 | 56.06 | 82.91 | 69.50 | 76.48 | 62.65 | 35.33 | 65.07 | 64.14 |
| RHO-1 ($\rho = 0.8$) | 65.28 | 52.14 | 79.62 | 56.57 | 83.71 | 71.21 | 75.85 | 63.03 | 35.33 | 65.23 | 64.80 |
| TOKENCLEANING ($\rho = 0.6$) | 65.58 | 51.35 | 79.70 | 55.89 | 82.49 | 70.02 | 76.56 | 62.34 | 38.56 | 65.17 | 64.77 |
| TOKENCLEANING ($\rho = 0.8$) | 66.42 | 51.87 | 79.66 | 55.97 | 83.08 | 69.37 | 76.56 | 62.33 | 38.56 | 64.86 | 64.87 |
| SSTOKEN ($\rho = 0.6$) | 64.07 | 52.90 | 79.88 | 56.57 | 83.63 | 71.94 | 75.85 | 62.62 | 38.71 | 65.46 | 65.16 |
| SSTOKEN ($\rho = 0.8$) | 66.55 | 52.96 | 79.97 | 56.22 | 82.49 | 71.94 | 76.24 | 62.98 | 42.70 | 66.30 | 65.84 |

## C.2 Detailed Performance with Adaptive $\rho$

In Sec. 4.4, we introduce a training-stage–based strategy that adaptively adjusts the token selection ratio $\rho$. Experiments on LLaMA-3.2-3B show that this dynamic schedule achieves consistently better performance than all manually specified fixed values of $\rho$ examined in our ablation study (i.e., $\rho \in 0, 0.2, 0.4, 0.6, 0.8, 1.0$). Table 6 reports the detailed results across the ten benchmark tasks.

Table 6: Comparison of ssToken with fixed and adaptive token selection ratios $\rho$ on LLaMA-3.2-3B.

| Methods | TriviaQA | TruthfulQA | MMLU | ARC-C | ARC-E | TyDiQA | Wino | HS | LogiQA | AGIEval | AVG |
|---|---|---|---|---|---|---|---|---|---|---|---|
| **Base model: LLaMA-3.2-3B** | | | | | | | | | | | |
| BASE ($\rho = 0$) | 50.85 | 39.37 | 56.20 | 42.15 | 74.49 | 19.05 | 69.06 | 55.21 | 29.49 | 30.04 | 46.59 |
| FULL ($\rho = 1$) | 52.50 | 43.97 | 57.34 | 45.56 | 76.42 | 39.48 | 70.56 | 55.75 | 29.80 | 32.11 | 50.35 |
| SSTOKEN ($\rho = 0.2$) | 51.70 | 53.82 | 55.63 | 44.20 | 73.65 | 47.50 | 70.48 | 54.32 | 33.03 | 32.91 | 51.72 |
| SSTOKEN ($\rho = 0.4$) | 51.79 | 51.62 | 56.22 | 45.65 | 75.55 | 50.70 | 70.80 | 55.73 | 31.80 | 33.47 | 52.33 |
| SSTOKEN ($\rho = 0.6$) | 54.04 | 47.42 | 57.02 | 45.82 | 76.42 | 51.04 | 70.17 | 56.19 | 32.87 | 33.97 | 52.50 |
| SSTOKEN ($\rho = 0.8$) | 53.99 | 45.72 | 57.49 | 45.65 | 77.57 | 50.10 | 70.24 | 56.26 | 31.95 | 33.88 | 52.29 |
| SSTOKEN (ADAPTIVE $\rho$) | 53.60 | 49.01 | 57.01 | 45.73 | 76.09 | 50.98 | 70.17 | 55.94 | 35.64 | 34.16 | **52.83** |

## C.3 Full-Parameter Fine-tuning Results

While main experiments in ssToken are performed under the LoRA fine-tuning regime to ensure consistency with prior token-level selection methods (Pang et al., 2025), the method itself is agnostic to the underlying parameterization strategy. ssToken depends only on token-level loss signals and deep-layer attention distributions, which remain identical regardless of whether the model is trained with LoRA adapters or via full-parameter updates. To verify this generality, we further conduct full-parameter supervised fine-tuning experiments on LLaMA-3.2-3B, evaluating ssToken alongside several representative baselines under matched training configurations.

For full-parameter SFT, we reduce the learning rate to 1e-5 to ensure stable optimization, while keeping all other hyperparameters consistent with the LoRA setting. The complete results are summarized in Table 7.

Table 7: Full-parameter fine-tuning results of LLaMA-3.2-3B across ten benchmarks. The best score in each column is highlighted in **bold**.

| Methods | TriviaQA | TruthfulQA | MMLU | ARC-C | ARC-E | TyDiQA | WinoGrande | HellaSwag | LogiQA | AGIEval | AVG |
|---|---|---|---|---|---|---|---|---|---|---|---|
| **Base model: LLaMA-3.2-3B** | | | | | | | | | | | |
| BASE | 50.85 | 39.37 | 56.20 | 42.15 | 74.49 | 19.05 | 69.06 | 55.21 | 29.49 | 30.04 | 46.59 |
| FULL-DATA | 51.11 | 40.25 | 56.54 | 43.60 | 76.81 | 48.40 | **70.48** | 55.71 | 30.64 | 32.73 | 50.63 |
| RANDOM | 50.90 | 41.22 | **56.90** | 43.11 | 76.54 | 45.26 | 70.17 | 55.22 | 31.41 | 31.68 | 50.24 |
| RHO-1 | 50.58 | 42.85 | 55.74 | 45.31 | **77.08** | 49.62 | 70.01 | 56.29 | 31.56 | 33.36 | 51.24 |
| TOKENCLEANING | 50.46 | 43.39 | 55.85 | 44.71 | 77.06 | 51.25 | 69.77 | **56.53** | **34.10** | 33.21 | 51.63 |
| SSTOKEN | **52.54** | **44.57** | 56.71 | **45.52** | 76.22 | **52.00** | 70.32 | 55.99 | 33.95 | **33.44** | **52.13** |

As shown in Table 1 and Table 7, ssToken consistently outperforms all baselines in both LoRA and full-parameter SFT settings, with highly similar improvement trends across the two fine-tuning regimes. This demonstrates that the effectiveness of ssToken is not tied to any specific parameterization strategy and remains stable across different training configurations.

## C.4 Multi-Epoch Training and EMA-Enhanced ssToken

To further evaluate the behavior of ssToken beyond the standard single-epoch SFT setting, we conduct additional experiments under multi-epoch supervision and examine the effects of the optional EMA-based history model update. These experiments are performed on LLaMA-3.2-3B using a reduced learning rate to avoid rapid overfitting. Some tasks (e.g., MMLU, ARC-Challenge, HellaSwag) converge rapidly within a single epoch and show limited gains in subsequent epochs. In contrast, other tasks exhibit steady improvements across multiple epochs, making them better suited for revealing the behavioral differences among token-selection strategies. We report results for three representative tasks (TruthfulQA, LogiQA, and TyDiQA) in Table 8.

Table 8: Effect of EMA for ssToken across epochs on TruthfulQA, LogiQA, and TyDiQA. The best score in each column for each dataset is highlighted in **bold**.

| Method | Base | Epoch 1 | Epoch 2 | Epoch 3 |
|---|---|---|---|---|
| **TruthfulQA** | | | | |
| Full-Data | 39.37 | 42.80 | 43.06 | 42.86 |
| Rho-1 | 39.37 | 42.62 | 43.17 | 43.50 |
| TokenCleaning | 39.37 | 42.89 | 43.46 | 44.67 |
| ssToken (w/o EMA) | 39.37 | **43.73** | 44.82 | 45.30 |
| ssToken (w/ EMA) | 39.37 | 43.41 | **45.13** | **45.72** |
| **LogiQA** | | | | |
| Full-Data | 29.49 | 29.19 | 30.57 | 31.03 |
| Rho-1 | 29.49 | 31.18 | 33.03 | 33.79 |
| TokenCleaning | 29.49 | 31.49 | 32.57 | 32.26 |
| ssToken (w/o EMA) | 29.49 | 33.49 | 34.10 | 34.25 |
| ssToken (w/ EMA) | 29.49 | **33.72** | **34.23** | **34.50** |
| **TyDiQA** | | | | |
| Full-Data | 19.05 | 30.27 | 35.33 | 35.12 |
| Rho-1 | 19.05 | 39.73 | 43.88 | 47.23 |
| TokenCleaning | 19.05 | 41.18 | **46.21** | 46.79 |
| ssToken (w/o EMA) | 19.05 | 41.83 | 45.72 | 47.50 |
| ssToken (w/ EMA) | 19.05 | **42.14** | 46.08 | **48.30** |

As shown in the results, for TruthfulQA, full-data SFT essentially saturates after the first epoch, whereas all token-selection methods continue to yield stable gains across additional epochs. Among them, ssToken consistently maintains the strongest performance across epochs.

Comparing ssToken with and without EMA, we observe that while the difference at epoch 1 is small, EMA consistently outperforms the non-EMA variant by epoch 3 across all three tasks. This suggests that, compared with using a fixed history model, an EMA-updated history model provides more stable and informative guidance during long-horizon training. These results further support the potential of ssToken in multi-epoch and continual-learning scenarios.

## D  ALGORITHMIC FRAMEWORK OF SSTOKEN

In this section, we summarize the overall algorithmic framework of ssToken in Algorithm 1, highlighting its two key components: self-modulated token selection and semantic-aware token importance estimation.

## E  VISUALIZATION OF REL DYNAMICS DURING TRAINING

To better understand how self-modulation emerges during optimization, we visualize the evolution of Retrospective Excess Loss (REL) throughout training. For each training batch, we compute the maximum REL value across the response tokens of every sample and then average these values across the batch. The resulting learning curves are shown in Fig. 4: Fig. 4a illustrates ssToken without EMA updates, and Fig. 4b depicts the EMA-enhanced variant.

As shown in the figure, the batch-averaged REL increases sharply within the first 100 training steps, after which it continues to rise more gradually with small fluctuations. This indicates that token-level losses quickly develop sufficient separation, allowing REL-based token selection to become effective at a very early stage of training.

Moreover, the presence of the attention-based token selection signal further stabilizes early-stage token selection, helping the model focus on instruction-relevant tokens before REL fully converges.

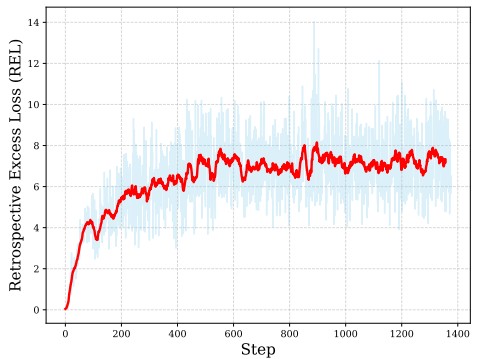

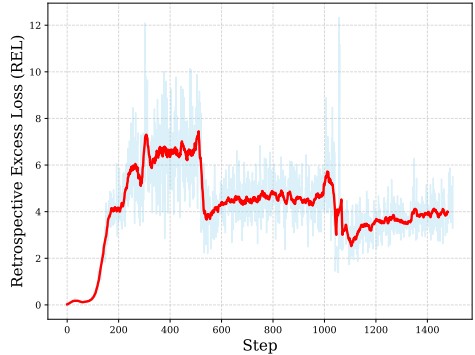

(a) REL dynamics of ssToken without EMA updating.

(b) REL dynamics of ssToken with EMA updating.

Figure 4: Visualization of the evolution of Retrospective Excess Loss (REL) during training

This complementary effect accelerates the emergence of meaningful REL values and contributes to the rapid performance improvement observed in the early training phase.

---

**Algorithm 1** ssToken: Self-modulated and Semantic-aware Token Selection

---

1: **Input:** Dataset $\mathcal{D}$, base model $\theta^0$, token selection ratio $\rho$, balance coefficient $\gamma$.
2: Initialize history model $\theta_{\text{his}} \leftarrow \theta^0$
3: **for** $t = 1$ **to** $T$ **do**
4:     Sample a sequence $\boldsymbol{x}$ from $\mathcal{D}$
5:     Compute $\text{REL}(x_i)$ for each token via Eq. 3
6:     Compute attention score $\text{Attn}(x_i)$ via Eq. 7
7:     Normalize $\text{REL}(x_i)$ via Eq. 8
8:     Calculate combined score:

$$\text{Score}(x_i) = \gamma \cdot \text{Normalize}(\text{REL}(x_i)) + (1 - \gamma) \cdot \text{AttnScore}(x_i)$$

9:     Select top-$\rho$ tokens by $\text{Score}(x_i)$
10:    Update $\theta^t$ with selected tokens
11:    **Optional:** Update $\theta_{\text{his}}$ (e.g., EMA, Eq. 4)
12: **end for**
13: **Return:** Fine-tuned model $\theta^t$

---

## F  CASE STUDIES OF TOKEN SELECTION

In this section, we present several case studies to illustrate the outcomes of different token selection strategies. For each sample, the tokens selected by a given method are highlighted in blue. In particular, for SSTOKEN, we additionally use orange to mark tokens that are primarily selected due to their high attention scores, distinguishing them from those selected mainly by loss-based signals. These visualizations provide an intuitive understanding of how various approaches behave and highlight the complementary role of semantic-aware signals in our method.

---

**Sample 1**

**User:** Write a response that appropriately completes the request.
### Instruction:
Name one type of organic compound.
### Response:

**Assistant (RHO-1):** One type of organic compound is an alkane, which is a saturated hydrocarbon that contains single bonds between carbon atoms.

**Assistant (TokenCleaning):** One type of organic compound is an alkane, which is a saturated hydrocarbon that contains single bonds between carbon atoms.

**Assistant (ssToken):** One type of organic compound is an alkane, which is a saturated hydrocarbon that contains single bonds between carbon atoms.

---

**Sample 2**

**User:** Provide instructions on how to accurately perform a bicep curl.

**Assistant (RHO-1):** 1 . Stand with your feet shoulder -width apart and grab a dumbbell in each hand with your palms facing forward . 2. Keep your elbows close to your body and slowly lift the weights towards your shoulders , while exhaling . Remember to keep your wrists straight and engage your biceps to lift the weight . 3. Pause for a second at the top of the curl , squeezing your b iceps . 4. Slowly lower the weights back down to the starting position, while inhaling . 5. Repeat the exercise for the desired number of reps. 6. To avoid injury , avoid swinging your arms or using momentum to lift the weights . Keep your core engaged and focus on isolating your b iceps for maximum benefit .

**Assistant (TokenCleaning):** 1 . Stand with your feet shoulder -width apart and grab a dumb bell in each hand with your palms facing forward . 2. Keep your elbows close to your body and slowly lift the weights towards your shoulders , while exhaling . Remember to keep your wrists straight and engage your biceps to lift the weight . 3. Pause for a second at the top of the curl , squeezing your b iceps . 4. Slowly lower the weights back down to the starting position, while inhaling . 5. Repeat the exercise for the desired number of reps. 6. To avoid injury , avoid swinging your arms or using momentum to lift the weights . Keep your core engaged and focus on isolating your b iceps for maximum benefit .

**Assistant (ssToken):** 1 . Stand with your feet shoulder-width apart and grab a dumbbell in each hand with your palms facing forward. 2 . Keep your elbows close to your body and slowly lift the weights towards your shoulders , while exh aling. Remember to keep your wrists straight and engage your b iceps to lift the weight. 3. Pause for a second at the top of the curl , squeezing your biceps. 4. Slow ly lower the weights back down to the starting position, while inhaling. 5. Repeat the exercise for the desired number of reps. 6 . To avoid injury , avoid swinging your arms or using momentum to lift the weights . Keep your core engaged and focus on isol ating your biceps for maximum benefit .

## Sample 3

**User:** In a restaurant kitchen, a chef uses a food processor to prepare a sauce. The recipe calls for 2 cups of chopped tomatoes, 1 cup of diced onions, and 1/2 cup of minced garlic. The food processor has a maximum capacity of 4 cups. If the chef wants to make enough sauce for 6 servings, how many times must the ingredients be processed in the food processor? Hint: Assume that the food processor can process the entire recipe at once.

**Assistant (RHO-1):** The recipe requires a total of 2 + 1 + 0.5 = 3.5 cups of ingredients. To make enough sauce for 6 servings, the chef needs to prepare 6 * 3.5 = 21 cups of ingredients in total. Since the food processor can process up to 4 cups of ingredients at a time, the chef needs to process the ingredients 21 / 4 = 5.25 times. However, since the chef cannot process ingredients a fraction of a time, they will need to process the ingredients 6 times to make enough sauce for 6 servings.

**Assistant (TokenCleaning):** The recipe requires a total of 2 + 1 + 0.5 = 3.5 cups of ingredients. To make enough sauce for 6 servings, the chef needs to prepare 6 * 3.5 = 21 cups of ingredients in total. Since the food processor can process up to 4 cups of ingredients at a time, the chef needs to process the ingredients 21 / 4 = 5.25 times. However, since the chef cannot process ingredients a fraction of a time, they will need to process the ingredients 6 times to make enough sauce for 6 servings.

**Assistant (ssToken):** The recipe requires a total of 2 + 1 + 0.5 = 3.5 cups of ingredients. To make enough sauce for 6 servings, the chef needs to prepare 6 * 3.5 = 21 cups of ingredients in total. Since the food processor can process up to 4 cups of ingredients at a time, the chef needs to process the ingredients 21 / 4 = 5.25 times. However, since the chef cannot process ingredients a fraction of a time, they will need to process the ingredients 6 times to make enough sauce for 6 servings.

## Sample 4

**User:** Using a deterministic algorithm, please generate a set of 3-6 integers, all of which must fall within the inclusive range of 10 to 100. These integers must satisfy the following conditions: the sum of all numbers in the set must be greater than or equal to 200, the difference between the largest and smallest numbers must be greater than or equal to 50, and no two numbers in the set may have a common factor greater than 1. Please provide the resulting set of integers.

**Assistant (RHO-1):** Here is a set of integers that satisfies the given conditions: - 13, 31, 97, 49 The sum of these numbers is 190, which is less than 200, so we can add another number to the set. - 11, 29, 89, 47, 35 This set satisfies all the given conditions. The sum of the numbers is 211, which is greater than or equal to 200. The difference between the largest and smallest numbers is 78, which is greater than or equal to 50. Finally, no two numbers in the set have a common factor greater than 1.

**Assistant (TokenCleaning):** Here is a set of integers that satisfies the given conditions: - 13, 31, 97, 49 The sum of these numbers is 190, which is less than 200, so we can add another number to the set. - 11, 29, 89, 47, 35 This set satisfies all the given conditions. The sum of the numbers is 211, which is greater than or equal to 200. The difference between the largest and smallest numbers is 78, which is greater than or equal to 50. Finally, no two numbers in the set have a common factor greater than 1.

**Assistant (ssToken):** Here is a set of integers that satisfies the given conditions: - 13, 31, 97, 49 The sum of these numbers is 190, which is less than 200, so we can add another number to the set. - 11, 29, 89, 47, 35 This set satisfies all the given conditions. The sum of the numbers is 211, which is greater than or equal to 200. The difference between the largest and smallest numbers is 78, which is greater than or equal to 50. Finally, no two numbers in the set have a common factor greater than 1.

