# OpenReview forum: "ssToken: Self-modulated and Semantic-aware Token Selection for LLM Fine-tuning"
_ICLR.cc/2026/Conference — ICLR 2026 Poster_

### Official Review · Reviewer_4Xmz · 2025-10-18

**Soundness:** 2
**Presentation:** 3
**Contribution:** 2
**Rating:** 6
**Confidence:** 3

**Summary:**

This paper proposes ssToken, a token-level data selection method for supervised fine-tuning (SFT) of large language models. The key contributions are: (1) a self-modulated selection strategy using retrospective excess loss (REL) that eliminates the need for training a separate reference model, and (2) a semantic-aware attention-based metric that complements loss-based selection. Experiments across models ranging from 3B to 14B parameters demonstrate improvements over full-data fine-tuning and prior token selection methods.

**Strengths:**

1. Well-motivated approach: The paper clearly identifies two practical limitations of existing token selection methods (reference model requirement and sole reliance on loss) and proposes targeted solutions.
2. Novel self-modulated paradigm: Using the history model instead of a separate reference model is elegant and practical. The formulation of REL is intuitive and eliminates additional training costs.
3. Complementary signals: The integration of attention-based semantic information with loss-based selection is well-motivated. The paper demonstrates that these signals are indeed orthogonal and mutually reinforcing.
4. Comprehensive experiments: Evaluation across multiple model families (LLaMA and Qwen), sizes (3B-14B), and 10 benchmarks provides good coverage. Ablation studies on γ and ρ are thorough.
5. Practical implementation: The lightweight attention extraction design compatible with FlashAttention is a thoughtful engineering contribution.
6. Consistent improvements: ssToken shows stable gains across different settings, which is valuable for practical adoption.

**Weaknesses:**

1. The reference models in baseline methods are trained on only 10k samples (20% of data), which may not provide fair comparison. A stronger reference trained on more data might close the gap.
2. The claim of "maintaining training efficiency" (Fig. 2) might be misleading since tokens still participate in forward pass. The computational overhead reduction is minimal.
3. The paper acknowledges that "history and current models share identical parameters at the beginning of training, which makes early-stage token selection nearly random" (line 210-212). However, there is no analysis of how many steps are needed before REL becomes effective, nor any experiments showing performance with different initialization strategies.

**Questions:**

1. Can you provide some analysis of when attention to prompt is vs. isn't a good indicator of token importance?
2. Can you show learning curves of REL values over training to demonstrate when self-modulation becomes effective?
3. Have you tried adaptive selection ratios (e.g., selecting different ρ per sample based on quality)?
4. What happens if you use ssToken for continual learning or multi-epoch training?

---

> ### Author Response · Authors · 2025-11-19
> **Part 1 of Rebuttal**
>
> Thank you for your positive and detailed assessment of our work. We sincerely appreciate your recognition of the clear motivation of our method, the novelty of the self-modulated REL formulation, the complementary nature of the attention-based semantic signal, the breadth and rigor of our experiments and ablations, the practicality of our lightweight implementation, and the consistent improvements achieved across models and benchmarks. We provide detailed responses to your questions below and hope that our explanations adequately address your concerns.
>
> **Q1: Clarification on using a 10k-sample reference model in baseline methods.**
>
> **A1:** Thank you for the comment. Our use of a 10k reference model strictly follows the official implementation of TokenCleaning. Importantly, these 10k samples are not randomly selected; as in TokenCleaning, they are chosen using a powerful sample-level selection method, DS² [1], to form a high-quality subset of the training data. In the TokenCleaning paper, a reference model trained on this 10k DS² subset already achieves performance close to full-data training.
>
> We also agree that using a stronger reference model might further improve baseline performance. However, this would require substantially more training for the reference model, thereby increasing the overall computational cost of reference-model–based methods such as RHO-1 and TokenCleaning.
>
> [1] Pang J et al. Improving Data Efficiency via Curating LLM-Driven Rating Systems. ICLR 2025.
>
> **Q2: Clarification on "maintaining training efficiency".**
>
> **A2:** We agree that token-level selection methods—including ours—do not reduce the computational cost of the forward pass, since all tokens still participate in forward computation. Their primary objective is to improve **training effectiveness**, rather than to shorten wall-clock training time. Nevertheless, compared with reference-model–based approaches, ssToken introduces only **marginal additional cost** during training.
>
> Following your advice, we have removed the phrase *“maintaining training efficiency”* from the main text. We now state more precisely that: *"Compared with the full-data baseline, ssToken delivers significant performance improvements with only a marginal increase in training time."*
>
> **Q3: When is prompt-based attention a good indicator of token importance, and when might it be less informative?**
>
> **A3:** Thank you for this insightful question. Below we summarize scenarios where prompt-based attention serves as a meaningful indicator of token importance, and cases where the signal may be less informative.
>
> **When it is a good indicator:** Prompt-based attention is most informative in instruction-following settings where response tokens are generated primarily by grounding on the user instruction. In such cases—e.g., QA, information extraction, reasoning over a given scenario—response tokens that strongly attend to the prompt typically correspond to semantically aligned or instruction-relevant content. Our results show that on instruction-oriented benchmarks such as TyDiQA, TriviaQA, and TruthfulQA, ssToken consistently outperforms both full-data SFT and loss-only baselines, suggesting that prompt attention provides a stable and meaningful signal for token importance in these settings.
>
> **When it may be less informative:** Prompt attention can be less reliable in tasks where generation depends primarily on intra-response dependencies rather than on the instruction itself—for example, free-form long-form generation or writing-oriented tasks, where later response tokens rely mostly on previously generated context. Additionally, complex reasoning tasks involve a chain of intermediate logical steps, and in such cases, focusing solely on prompt attention may appear insufficient for capturing token importance contributed by earlier reasoning steps. However, as shown in Table 1, ssToken consistently achieves superior performance on several reasoning-oriented benchmarks—such as LogiQA (formal logic and deductive reasoning) and AGIEval (exam-style questions involving multi-step reasoning). This demonstrates that, even though our metric uses only response→prompt attention, the introduction of semantic-aware signals still enables ssToken to outperform loss-only methods on tasks that require substantial reasoning capability.
>
> Importantly, our method uses deep-layer attention, where heads typically encode high-level semantic dependencies rather than syntactic or positional patterns. As shown in our ablations (see A3 for Reviewer P6EM), deep-layer attention yields stable token-importance scores across models and tasks and aligns well with performance improvements.
>
> Taken together, prompt-based attention is a reliable token-importance signal whenever the task requires grounding on the instruction—which includes the majority of modern SFT datasets—and our experiments validate that it remains robust even on benchmarks that involve substantial reasoning.

---

> ### Author Response · Authors · 2025-11-19
> **Part 2 of Rebuttal**
>
> **Q4: When does REL-based self-modulation become effective during training?**
>
> **A4:** Thank you for the suggestion. We agree that visualizing the evolution of REL provides clearer insight into how self-modulation becomes effective during training. We have added the corresponding learning curve in **Figure 4 of the Appendix**.
>
> As shown in the figure, the batch-averaged REL **increases sharply within the first ~100 training steps**, after which it continues to rise more gradually with small fluctuations. This indicates that token-level losses quickly develop sufficient separation, allowing REL-based token selection to become effective at a very early stage of training.
>
> Moreover, the presence of the attention-based token selection signal **further stabilizes early-stage token selection, helping the model focus on instruction-relevant tokens before REL fully converges**. This complementary effect accelerates the emergence of meaningful REL values and contributes to the rapid performance improvement in the early training phase.
>
>
> **Q5: On the adaptive token selection ratio (ρ)**
>
> **A5:** Thank you for this valuable suggestion. We first clarify that manually setting a fixed token selection ratio (ρ) is a common practice in existing loss-based token selection methods such as RHO-1 and TokenCleaning, all of which adopt a fixed ρ. Since the main contributions of our paper focus on addressing their two fundamental limitations— (1) requiring training or accessing an additional reference model, and  (2) relying solely on loss as the selection signal—  we followed the same fixed-ρ setting in the main paper to ensure a fair comparison and maintain a focused scope.
>
> We fully agree with the reviewer that keeping ρ fixed throughout training may not adequately reflect the model’s evolving capability. In the main text, we already provided extensive ablations and explicitly pointed out in the *Limitations* section that adaptive ρ is a promising direction. Motivated by your suggestion, we further implemented and evaluated a simple **global adaptive schedule** during the rebuttal period, allowing early-stage and late-stage token selection to behave more appropriately.
>
> Concretely, we define ρ as a monotonically decreasing function of training progress:
>
> - **Early stage (unstable model predictions):**
>   Use a larger ρ (e.g., ~0.8) to retain more tokens and reduce noise caused by unstable REL signals.
> - **Late stage (model becomes more reliable):**
>   Gradually decrease ρ (e.g., down to 0.4–0.5), so that token selection focuses more strongly on truly informative tokens.
>
> For simplicity and efficiency, we adopt a training-step-based decay schedule (linear or cosine), which introduces **no additional computational cost**. The schedule we used is:
>
> $$
> \rho_t = \rho_{\min} + (\rho_{\max} - \rho_{\min}) \left(1 - \frac{t}{T}\right)^\beta.
> $$
>
> On LLaMA-3.2-3B, we observed that the adaptive schedule brings **additional performance improvements** over the fixed-ρ baseline.
>
> | **Methods** | **TriviaQA** | **TruthfulQA** | **MMLU** | **ARC-C** | **ARC-E** | **TyDiQA** | **WinoGrande** | **HellaSwag** | **LogiQA** | **AGIEval** | **AVG** |
> |-|-|-|-|-|-|-|-|-|-|-|-|
> |Base|50.85|39.37| 56.20 | 42.15 | 74.49 | 19.05 | 69.06 | 55.21 | 29.49 | 30.04 | 46.59 |
> | Full | 52.50 | 43.97 | **57.34** | 45.56 | 76.42 | 39.48 | 70.56 | 55.75 | 29.80 | 32.11 | 50.35 |
> | Random | 52.71 | 43.72 | 56.84 | 45.14 | 76.60 | 35.66 | 70.03 | 55.70 | 29.95 | 31.89 | 49.82 |
> | Rho-1 | 53.14 | 44.35 | 56.78 | 45.31 | 76.98 | 50.47 | 70.17 | 56.67 | 29.80 | 32.79 | 51.65 |
> | TokenCleaning | 53.09 | 44.79 | 57.17 | 45.22 | **77.06** | **51.40** | 69.93 | **56.69** | 30.57 | 32.81 | 51.87 |
> | **ssToken** | **54.04** | 47.42 | 57.02 | **45.82** | 76.42 | 51.04 | **70.56** | 56.19 | 32.87 | 33.97 | 52.50 |
> | **ssToken (adaptive ρ)** | 53.60 | **49.01** | 57.01 | 45.73 | 76.09 | 50.98 | 70.17 | 55.94 | **35.64** | **34.16** | **52.83** |
>
> | Methods | Base (ρ=0) | Full (ρ=1) | ssToken (ρ=0.2) | ssToken (ρ=0.4) | ssToken (ρ=0.6) | ssToken (ρ=0.8) | ssToken (adaptive ρ) |
> |-|-|-|-|-|-|-|-|
> | Average Performance| 46.59| 50.35| 51.72|52.33| 52.50|52.29|**52.83**|
>
>
>
> We believe the improvement mainly comes from:
>
> 1. **Early stage:** the model has not yet formed stable estimations of token importance, so a larger ρ prevents mistakenly discarding valuable tokens;
> 2. **Late stage:** the model becomes more capable of identifying informative tokens, and a smaller ρ strengthens this focus, improving final performance;
> 3. **Dynamic ρ effectively forms an online curriculum**, allowing token selection to gradually shrink as the model becomes stronger.
>
> In addition to scheduling ρ based on training progress, assigning sample-specific ρ values according to sample quality is also a feasible strategy. The improvements observed with our dynamic schedule further confirm that adaptive ρ is an unexplored yet highly promising direction for token-level selection methods.

---

> ### Author Response · Authors · 2025-11-19
> **Part 3 of Rebuttal**
>
> **Q6: How does ssToken perform under multi-epoch or continual-learning settings?**
>
> **A6:** Thank you for this thoughtful question. To examine how ssToken behaves under multi-epoch or continual-learning settings, we conducted additional experiments across several token-selection methods. For stability, we used a smaller learning rate to avoid rapid overfitting.  Some tasks (e.g., MMLU, ARC-Challenge, HellaSwag) converge rapidly within a single epoch and show limited gains in subsequent epochs. In contrast, other tasks exhibit steady improvements across multiple epochs, making them better suited for revealing the behavioral differences among token-selection strategies. We report results for three representative tasks—TruthfulQA, LogiQA, and TyDiQA—in the tables below.
>
> **TruthfulQA Results**
> | **Method**        | **Base** | **Epoch 1** | **Epoch 2** | **Epoch 3** |
> |-------------------|----------|-------------|-------------|-------------|
> | Full-Data         | 39.37    | 42.80       | 43.06       | 42.86       |
> | Rho-1             | 39.37    | 42.62       | 43.17       | 43.50       |
> | TokenCleaning     | 39.37    | 42.89       | 43.46       | 44.67       |
> | ssToken (w/o EMA)          | 39.37    | **43.73**       | 44.82       | 45.30       |
> | ssToken (w/ EMA)     | 39.37    | 43.41       | **45.13**       | **45.72**       |
>
>
> **LogiQA Results**
> | **Method**        | **Base** | **Epoch 1** | **Epoch 2** | **Epoch 3** |
> |-------------------|----------|-------------|-------------|-------------|
> | Full-Data         | 29.49    | 29.19       | 30.57       | 31.03       |
> | Rho-1             | 29.49    | 31.18       | 33.03       | 33.79       |
> | TokenCleaning     | 29.49    | 31.49       | 32.57       | 32.26       |
> | ssToken (w/o EMA)    | 29.49    | 33.49       | 34.10       | 34.25       |
> | ssToken (w/ EMA)     | 29.49    | **33.72**       | **34.23**       | **34.50**       |
>
> **TyDiQA Results**
> | **Method**        | **Base** | **Epoch 1** | **Epoch 2** | **Epoch 3** |
> |-------------------|----------|-------------|-------------|-------------|
> | Full-Data         | 19.05    | 30.27       | 35.33       | 35.12       |
> | Rho-1             | 19.05    | 39.73       | 43.88       | 47.23       |
> | TokenCleaning     | 19.05    | 41.18       | **46.21**       | 46.79       |
> | ssToken (w/o EMA)    | 19.05    | 41.83       | 45.72       | 47.50       |
> | ssToken (w/ EMA)     | 19.05    | **42.14**       | 46.08       | **48.30**       |
>
>
> In addition, to further evaluate the effectiveness of the optional EMA mechanism in ssToken for long-horizon training, we also include results for ssToken (w/ EMA). We use an EMA smoothing factor of α = 0.8 and update the history model every 0.5 epoch (and we also visualize the REL learning curves under the EMA setting in Figure 4 of the Appendix). As shown in the results, for TruthfulQA, full-data SFT essentially saturates after the first epoch, whereas all token-selection methods continue to yield stable gains across additional epochs. Among them, **ssToken consistently maintains the strongest performance across epochs.**
>
> Comparing ssToken with and without EMA, we observe that while the difference at epoch 1 is small, **EMA consistently outperforms the non-EMA variant by epoch 3 across all three tasks**. This suggests that, compared with using a fixed history model, an EMA-updated history model provides more stable and informative guidance during long-horizon training. These results further support the potential of ssToken in multi-epoch and continual-learning scenarios.
>
> ### **Overall summary:**
> We have incorporated all relevant points from the rebuttal into the revised manuscript. Specifically, the REL learning curves of ssToken (and its EMA-enhanced variant) have been visualized in Figure4 of the Appendix and analyzed in Appendix E; the experiments and analyses on the adaptive token selection ratio $\rho$ have been added to Sec.4.4 and Appendix C.2; and the performance and analysis of ssToken under multi-epoch and long-horizon training have been included in Appendix C.4.
>
> These additions further strengthen the completeness and clarity of the paper. We sincerely appreciate your careful review and valuable feedback, and we hope that our revisions and responses adequately address your concerns. Should you have any further questions, we would be glad to discuss them.

---

### Official Review · Reviewer_RRS3 · 2025-10-26

**Soundness:** 2
**Presentation:** 3
**Contribution:** 2
**Rating:** 4
**Confidence:** 4

**Summary:**

The paper proposes ssToken, a token-level data selection method for SFT that (i) removes the need for an externally trained or stronger reference model and (ii) augments loss-based selection with a lightweight semantic-aware signal derived from attention.

Instead of computing “excess loss” against a fixed reference model, the authors consider Retrospective Excess Loss (REL) computed against a history model, which is futher combined with attention-based token importance, another measurement of token weight by summing up attention scores over all prompt tokens. Lastly they consider the convex combination of the two signals and only train on top-x fraction of response tokens.

Empirically, they showed ssToken outperforms both full-data SFT and prior token selection baselines across various base models and benchmarks.

**Strengths:**

1, The paper proposed an innovative prompt-focused attention metric
2, Details are well discussed, such attention-layer’s preference, hooks+recompute approach for FlashAttention compatibility,
3, The paper is well structured (motivation → method → ablations → results) with comprehensive figures and explicit algorithmic details
4, The topic of token-level data selection is an increasingly important topic for better data-efficiency. The methods introduced in the paper are easy to adapt and implement.

**Weaknesses:**

1, While the author claims the attention based measurement is lightweight and no reference training is needed, it still helps to quantify the additional costs of 1) forward passes to compute REL using history model (2) attention recomputation, and especially when scaled to larger models and longer contexts
2, The paper notes that if history agrees with the current model at the beginning then the selection is “nearly random”, and suggests EMA, but eventually admits that EMA doesn’t yield gain and leads to additional overhead. What’s the take away on this discussion? The authors are expected to explain why this is the case, and whether this is due to small SFT. Does adaptive history update help on longer training horizons or larger corpora?
3, The attention-based approach focuses exclusively on attention from prompt, and overlooks contributions from other response tokens. This omission can be problematic in reasoning-heavy tasks such as mathematical problem solving, where the model must follow a chain of logic across intermediate steps. In such cases, the next step often depends more on information contained in previous reasoning steps than on the original prompt. Ignoring attention flows among response tokens may therefore hurt model’s ability to capture these intra-response dependencies.
4, A more concerning point is that all experiments were performed with LoRA, which is a non-standard approach for fine-tuning LLMs. This leaves the audience wondering if the method works for full tuning and or even pretraining.

**Questions:**

1, How many extra forwards/backwards per step are required to compute REL and the attention metric? Is REL computed with an additional pass through the history model each step/batch, or are there cached losses?
2, You mention an optional EMA but ultimately fix the base model as history due to small data. For larger/longer SFT, do you expect EMA to stabilize selection and reduce early randomness? Can you share a small-scale ablation to illustrate the trend?
3, In prompt based attention approach, what happens if the response context is long? Will position bias become a problem? Does your analysis cover this regime?
4, Which exact layer index do you use across models? Do model families disagree on which depth is best?
5, Have you considered head-weighted aggregation (e.g., entropy-based or gradient-based head importance) rather than simple averaging?
6, can you justify why omitting response tokens' contributions in the attention approach?
7, Do the conclusions change for full-parameter SFT vs LoRA? Can you add ablation studies to support this?

---

> ### Author Response · Authors · 2025-11-19
> **Part 1 of Rebuttal**
>
> Thank you very much for your careful and valuable comments. We appreciate the opportunity to address your concerns, and we provide detailed responses to each of your questions below.
>
> **Q1: How many extra forwards/backwards per step are required to compute REL and the attention metric? Is REL computed with an additional pass through the history model each step/batch, or are there cached losses?**
>
> **A1:** Thank you for the question. We first clarify that **ssToken does not require any additional forward or backward passes during training, nor does it run the history model at every step**. REL does not rely on extra inference during SFT. Instead, we follow a lightweight two-stage implementation that avoids loading two models simultaneously and keeps the training loop almost identical to standard full-data SFT.
>
> Before SFT begins, we run the history model (i.e., the base LLM before fine-tuning) over the training corpus once in an offline preprocessing stage, storing only the token-level losses. This step is inference-only, requires no gradients, and is very fast (only a few minutes). During fine-tuning, the current model performs a single normal forward pass to compute the token-level losses. The REL values are then obtained simply by **subtracting the cached history losses**, which adds no extra forward or backward computation. As a result, REL is computed per token and per step automatically within the same forward computation of the current model, without requiring any additional passes or model loading.
>
> For the attention-based metric, our implementation is fully compatible with FlashAttention. During the forward pass, we register a lightweight hook to cache the hidden states of a single target layer, and after the forward computation finishes, we recompute only that layer once to extract its attention matrix. This design avoids outputting attention matrices throughout the forward pass and preserves full compatibility with FlashAttention. Since a full forward pass also includes the FFN computation of all layers, the additional overhead is approximately less than 1/28 of a full forward pass in LLaMA-3.2-3B (28 layers), and even smaller in larger models.
>
> To provide a more realistic assessment of overhead, we compare the **end-to-end total training time over the full dataset** under identical batch size, sequence length, and number of training steps (i.e. Figure 2 in the main paper).
>
> | Total training time (minutes) | 3B   | 8B   | 14B  |
> |-------------------------------|------|------|------|
> | Full-data SFT                 | 20.8 | 44.1 | 87.2 |
> | ssToken                 | 23.5 | 47.9 | 93.9 |
>
>
> All measurements were obtained using 8× NVIDIA H200 GPUs. As shown, ssToken is only slightly slower than full-data fine-tuning, even after accounting for the one-time history-model inference. In contrast, RHO-1 and TokenCleaning require training an additional reference model, and their total training time—reported in Figure 2—significantly exceeds that of both full-data SFT and ssToken.

---

> ### Author Response · Authors · 2025-11-19
> **Part 2 of Rebuttal**
>
> **Q2: How does EMA affect early-stage behavior and long-horizon training in ssToken?**
>
> **A2:** As discussed in the paper, the motivation for introducing EMA is that, unlike reference-model–based methods, ssToken’s history model can be updated at small additional cost and therefore provides more stable guidance in long-horizon training. Since Reviewer 4Xmz also asked about multi-epoch and longer SFT behavior, we enabled the EMA mechanism in this setting and compared different token-selection methods across multiple epochs (using a smaller learning rate to avoid rapid overfitting). Some tasks (e.g., MMLU, ARC-Challenge, HellaSwag) converge rapidly within a single epoch with limited subsequent improvement. In contrast, some other tasks continue to improve across multiple epochs, which makes them suitable for revealing differences among token-selection methods. Below, we report detailed results for TruthfulQA, LogiQA, and TyDiQA as representative examples.
>
> **TruthfulQA Results**
> | **Method**        | **Base** | **Epoch 1** | **Epoch 2** | **Epoch 3** |
> |-------------------|----------|-------------|-------------|-------------|
> | Full-Data         | 39.37    | 42.80       | 43.06       | 42.86       |
> | Rho-1             | 39.37    | 42.62       | 43.17       | 43.50       |
> | TokenCleaning     | 39.37    | 42.89       | 43.46       | 44.67       |
> | ssToken (w/o EMA)          | 39.37    | **43.73**       | 44.82       | 45.30       |
> | ssToken (w/ EMA)     | 39.37    | 43.41       | **45.13**       | **45.72**       |
>
>
> **LogiQA Results**
> | **Method**        | **Base** | **Epoch 1** | **Epoch 2** | **Epoch 3** |
> |-------------------|----------|-------------|-------------|-------------|
> | Full-Data         | 29.49    | 29.19       | 30.57       | 31.03       |
> | Rho-1             | 29.49    | 31.18       | 33.03       | 33.79       |
> | TokenCleaning     | 29.49    | 31.49       | 32.57       | 32.26       |
> | ssToken (w/o EMA)    | 29.49    | 33.49       | 34.10       | 34.25       |
> | ssToken (w/ EMA)     | 29.49    | **33.72**       | **34.23**       | **34.50**       |
>
> **TyDiQA Results**
> | **Method**        | **Base** | **Epoch 1** | **Epoch 2** | **Epoch 3** |
> |-------------------|----------|-------------|-------------|-------------|
> | Full-Data         | 19.05    | 30.27       | 35.33       | 35.12       |
> | Rho-1             | 19.05    | 39.73       | 43.88       | 47.23       |
> | TokenCleaning     | 19.05    | 41.18       | **46.21**       | 46.79       |
> | ssToken (w/o EMA)    | 19.05    | 41.83       | 45.72       | 47.50       |
> | ssToken (w/ EMA)     | 19.05    | **42.14**       | 46.08       | **48.30**       |
>
> In our experiments, we use an EMA smoothing factor of α = 0.8, applying EMA updates every 0.5 epoch. As shown in the tables, full-data SFT saturates after the first epoch on TruthfulQA, whereas all token-selection methods continue to provide steady gains with additional epochs. Among them, **ssToken consistently maintains the strongest performance**.
>
> Comparing ssToken with and without EMA, we observe that while the difference at epoch 1 is small, **EMA consistently outperforms the non-EMA variant by epoch 3 across all three tasks**. This indicates that, compared with using a fixed history model, an EMA-updated history model provides more stable and up-to-date guidance during long-horizon training.
>
> Regarding early-stage “randomness”: EMA does not reduce it, because the current and history models are identical at the very beginning of training. However, As suggested by Reviewer 4Xmz, we have also visualized the REL learning curves of ssToken and its EMA-enhanced variant in Figures 4  of the Appendix. As shown in the figure, the batch-averaged REL **increases sharply within the first ~100 training steps**, after which it continues to rise more gradually with small fluctuations. This indicates that token-level losses quickly develop sufficient separation, allowing REL-based token selection to become effective at a very early stage of training.
>
> Moreover, the presence of the attention-based token selection signal **further stabilizes early-stage token selection, helping the model focus on instruction-relevant tokens before REL fully converges**. This complementary effect accelerates the emergence of meaningful REL values and contributes to the rapid performance improvement observed in the early training phase.
>
> In summary, the main role of EMA is to provide a more stable and up-to-date history model for long-horizon training. The early-stage randomness naturally decays within the first few training steps, and the attention-based signal can offer reliable token-selection guidance before REL becomes fully effective.

---

> ### Author Response · Authors · 2025-11-19
> **Part 3 of Rebuttal**
>
> **Q3: Which specific layer index is used across different models? Do different model families disagree on which depth is optimal?**
>
> **A3:** As stated in the main paper, we use **deep-layer attention (i.e., the final transformer layer)** for all model families. The exact layer indices for each model are:
>
> | Model              | Total Layers | Selected Layer (Deep) |
> |--------------------|--------------|------------------------|
> | LLaMA-3.2-3B       | 28           | 28                     |
> | LLaMA-3.1-8B       | 32           | 32                     |
> | Qwen-2.5-7B        | 28           | 28                     |
> | Qwen-2.5-14B       | 48           | 48                     |
>
> In the original Appendix, we have already conducted a systematic layer-selection ablation on LLaMA-3.2-3B. To further investigate whether different model families prefer different depths, we performed the same ablation on Qwen-2.5-7B during the rebuttal period. The full results for both models are included in Sec. 4.3 in the revised main text.
>
>
> | **LLaMA-3.2-3B** | **TriviaQA** | **TruthfulQA** | **MMLU** | **ARC-C** | **ARC-E** | **TyDiQA** | **WinoGrande** | **HellaSwag** | **LogiQA** | **AGIEval** | **AVG** |
> |------------|--------------|----------------|----------|-----------|-----------|------------|----------|--------|------------|-------------|---------|
> | **Shallow Attention (layer 1)** | 54.36 | 45.12 | 56.23 | 45.48 | **77.48** | 48.52 | 70.09 | 53.76 | 31.80 | 33.43 | 51.63 |
> | **Medium Attention (layer 14)** | **54.45** | 47.04 | **57.43** | 45.73 | 77.23 | 49.20 | 69.77 | 55.40 | 29.95 | 32.96 | 51.92 |
> | **Deep Attention (layer 28)** | 54.04 | **47.42** | 57.02 | **45.82** | 76.42 | **51.04** | **70.17** | **56.19** | **32.87** | **33.97** | **52.50** |
>
>
>
> | **Qwen-2.5-7B** | **TriviaQA** | **TruthfulQA** | **MMLU** | **ARC-C** | **ARC-E** | **TyDiQA** | **WinoGrande** | **HellaSwag** | **LogiQA** | **AGIEval** | **AVG** |
> |------------------|--------------|----------------|----------|-----------|-----------|------------|----------------|---------------|------------|-------------|---------|
> | **Shallow Attention (layer 1)** | 55.47 | 46.25 | **74.26** | 49.23 | 81.10 | 68.70 | 72.69 | 59.32 | 27.90 | 57.06 | 59.20 |
> | **Medium Attention (layer 14)** | **57.20** | 48.40 | 73.78 | **51.47** | **82.58** | 68.94 | 72.74 | 58.31 | 29.62 | 55.26 | 59.83 |
> | **Deep Attention (layer 28)** | 56.73 | **50.75** | 74.18 | 50.51 | 81.48 | **69.74** | **72.93** | **60.30** | **31.03** | **57.18** | **60.48** |
>
>
> Across both model families, several consistent patterns emerge. First, for knowledge-intensive benchmarks that rely on pretraining (e.g., MMLU), different attention layers do not exhibit a strictly dominant winner. In contrast, for QA-style tasks (e.g., TruthfulQA, TyDiQA) and logical/commonsense reasoning benchmarks (e.g., LogiQA, HellaSwag), **deep-layer attention consistently performs best**.
>
> More concretely, on LLaMA-3.2-3B, deep-layer attention achieves the best score on 7 out of 10 tasks, and on Qwen-2.5-7B, it achieves the best score on 6 out of 10 tasks. In both model families, **the deep-layer variant also obtains the highest overall AVG score, indicating the most stable and generalizable performance**.
>
> As discussed in the main paper, this trend is also aligned with findings from interpretability and probing literature: shallow layers predominantly encode local syntactic or positional patterns, whereas deeper layers capture more abstract semantic structure and task-relevant long-range dependencies. Importantly, this pattern remains consistent not only across different model sizes but also across **different model families (LLaMA vs. Qwen)**, confirming the robustness of choosing deep-layer attention as our semantic-aware signal.

---

> ### Author Response · Authors · 2025-11-19
> **Part 4 of Rebuttal**
>
> **Q4: Clarification on (1) the design choice of prompt-based attention and (2) its behavior under long-context settings.**
>
> **A4:** We first explain why we do not use response→response attention in our method. Our reasoning is based on two main considerations:
>
> (1) We acknowledge that response→response attention is also meaningful: the extent to which later tokens attend to an earlier token can indeed reflect that token’s importance within the response. However, **under the causal mask of autoregressive transformers, estimating token importance using “the sum of attentions from all other response tokens to the target token” introduces a inherent position bias**. The first response token receives attention from all subsequent tokens, whereas the last response token can only attend to itself. This discrepancy is structural rather than semantic, and is why such estimation is more suitable for BERT-style bidirectional models. Designing a principled way to leverage response→response attention under causal masking requires more sophisticated mechanisms, which we view as a future direction.
>
> (2) In contrast, **response→prompt attention has a clearer and more intuitive interpretation in the SFT setting**. The prompt length is fixed, and every response token can attend to the same set of prompt tokens. Thus, defining a token’s importance as its cumulative attention to the prompt naturally alleviates position bias. As shown in the case studies in Appendix F, tokens selected by attention score are distributed approximately uniformly across the response. Moreover, the core goal of SFT is to improve instruction-following through instruction tuning, and response→prompt attention directly captures whether a token is generated with reference to the instruction. This design naturally aligns with SFT.
>
> We acknowledge that the logical flow among response tokens can indeed be helpful for complex reasoning. However, as shown in Table 1, ssToken achieves superior performance on several reasoning-oriented benchmarks—such as LogiQA (formal logic and deductive reasoning) and AGIEval (exam-style questions involving multi-step reasoning). This demonstrates that, even though our metric uses only response→prompt attention, the introduction of semantic-aware signals still enables ssToken to outperform loss-only methods such as TokenCleaning on tasks that require substantial reasoning capability.
>
> Regarding long-context scenarios, while later response tokens may assign slightly lower attention to the prompt, we did not observe severe or systematic position bias in practice; many later tokens still receive high attention scores.
>
> In summary, response→response attention is valuable but currently difficult to use directly due to strong position bias under causal masking. In contrast, response→prompt attention effectively mitigates this bias, aligns naturally with the goals of SFT, and yields consistently strong empirical performance—hence our choice.
>
>
> **Q5: Why not use head-weighted aggregation (entropy-based / gradient-based) instead of simple averaging?**
>
> **A5:** Thank you for the suggestion. We agree that different attention heads in Transformers may carry different functional roles, and some heads may encode more semantic information. This makes weighted-sum aggregation a plausible direction for further improving attention-based importance estimation. However, compared with simple averaging, identifying head importance and applying head-level weighting is often more challenging to ensure stability, and it introduces additional computational and implementation overhead. For example, gradient-based importance requires computing or caching extra gradients. These additional costs conflict with the core design goals of ssToken—**lightweight, stable, reproducible, and plug-and-play**.
>
> In summary, we choose simple averaging not merely for its computational efficiency, but based on a comprehensive consideration of stability, generalization, and empirical performance. It yields highly consistent results across different model scales, model families, and task types, making it a reliable and robust choice in practice. Nonetheless, we agree that more sophisticated head-weighted aggregation remains an interesting direction for future work and may further improve token selection in specialized scenarios.

---

> ### Author Response · Authors · 2025-11-19
> **Part 5 of Rebuttal**
>
> **Q6: Does ssToken remain effective under full-parameter fine-tuning, and how does it compare to LoRA-based SFT?**
>
> **A:** Thank you for raising this important point. We use LoRA for fine-tuning primarily to **follow the exact experimental setup of the TokenCleaning paper**, ensuring a fair comparison across all baselines. Moreover, after LoRA training, we explicitly perform a merge step so that the LoRA updates are fused back into the base model parameters. As a result, the final fine-tuned model has the same parameter shape as the original base model. It is also important to clarify that **the core idea of ssToken is entirely independent of the fine-tuning parameterization**. The method relies only on token-level loss and deep-layer attention signals, which are fully decoupled from whether the model is trained via LoRA or full-parameter updates.
>
> We highly appreciate the reviewer’s suggestion. To further verify that ssToken is applicable across different fine-tuning regimes, we followed your recommendation and conducted **full-parameter SFT experiments** on LLaMA-3.2-3B, comparing several token selection methods under exactly matched training configurations. Our [anonymous codebase](https://anonymous.4open.science/r/Submission2116-B7C5) naturally supports full-parameter SFT—one only needs to remove the `--use_lora` flag and the corresponding LoRA arguments in `finetune.sh`. For full-parameter SFT, we used a smaller learning rate to ensure stable training. The complete results are shown below:
>
> **LORA Fine-tuning**
> | **Methods** | **TriviaQA** | **TruthfulQA** | **MMLU** | **ARC-C** | **ARC-E** | **TyDiQA** | **WinoGrande** | **HellaSwag** | **LogiQA** | **AGIEval** | **AVG** |
> |-------------|-------------|----------------|----------|-----------|-----------|------------|----------|--------|------------|-------------|---------|
> | Base | 50.85 | 39.37 | 56.20 | 42.15 | 74.49 | 19.05 | 69.06 | 55.21 | 29.49 | 30.04 | 46.59 |
> | Full-Data | 52.50 | 43.97 | **57.34** | 45.56 | 76.42 | 39.48 | 70.56 | 55.75 | 29.80 | 32.11 | 50.35 |
> | Random | 52.71 | 43.72 | 56.84 | 45.14 | 76.60 | 35.66 | 70.03 | 55.70 | 29.95 | 31.89 | 49.82 |
> | Rho-1 | 53.14 | 44.35 | 56.78 | 45.31 | 76.98 | 50.47 | 70.17 | 56.67 | 29.80 | 32.79 | 51.65 |
> | TokenCleaning | 53.09 | 44.79 | 57.17 | 45.22 | **77.06** | **51.40** | 69.93 | **56.69** | 30.57 | 32.81 | 51.87 |
> | **ssToken** | **54.04** | **47.42** | 57.02 | **45.82** | 76.42 | 51.04 | **70.56** | 56.19 | **32.87** | **33.97** | **52.50** |
>
>
> **Full-Parameter Fine-tuning**
> | **Methods** | **TriviaQA** | **TruthfulQA** | **MMLU** | **ARC-C** | **ARC-E** | **TyDiQA** | **WinoGrande** | **HellaSwag** | **LogiQA** | **AGIEval** | **AVG** |
> |-------------|-------------|----------------|----------|-----------|-----------|------------|----------------|--------------|------------|-------------|---------|
> | Base | 50.85 | 39.37 | 56.20 | 42.15 | 74.49 | 19.05 | 69.06 | 55.21 | 29.49 | 30.04 | 46.59 |
> | Full-Data | 51.11 | 40.25 | 56.54 | 43.60 | 76.81 | 48.40 | **70.48** | 55.71 | 30.64 | 32.73 | 50.63 |
> | Random | 50.90 | 41.22 | **56.90** | 43.11 | 76.54 | 45.26 | 70.17 | 55.22 | 31.41 | 31.68 | 50.24 |
> | Rho-1 | 50.58 | 42.85 | 55.74 | 45.31 | **77.08** | 49.62 | 70.01 | 56.29 | 31.56 | 33.36 | 51.24 |
> | TokenCleaning | 50.46 | 43.39 | 55.85 | 44.71 | 77.06 | 51.25 | 69.77 | **56.53** | **34.10** | 33.21 | 51.63 |
> | ssToken | **52.54** | **44.57** | 56.71 | **45.52** | 76.22 | **52.00** | 70.32 | 55.99 | 33.95 | **33.44** | **52.13** |
>
>
>
> As shown in the table, **ssToken consistently outperforms all baselines in both LoRA and full-parameter SFT settings**, with highly similar improvement trends across the two fine-tuning regimes. This demonstrates that the effectiveness of ssToken is not tied to any specific parameterization strategy and remains stable across different training configurations.
>
> ### **Overall summary:**
> We have incorporated all relevant points from the rebuttal into the revised manuscript. Specifically, the detailed computation procedure of REL has been clarified in line 795–800; the performance and analysis of ssToken (and its EMA-enhanced variant) under multi-epoch and long-horizon training have been added to Appendix C.4; the ablation and discussion on attention-layer choice have been moved from the Appendix to Sec. 4.3, together with newly added results on Qwen-2.5-7B; and the experimental results and analysis of different methods under full-parameter fine-tuning have been added to Appendix C.3.
>
> These additions further improve the completeness and clarity of the paper. We sincerely appreciate your careful review and valuable feedback, and we hope that our revisions and responses adequately address your concerns. Should you have any additional questions, we would be glad to discuss them.

---

> ### Author Response · Authors · 2025-11-27
>
> Dear Reviewer `RRS3`,
>
> I hope this message finds you well. We have provided detailed responses and clarifications to each of the weaknesses and questions you raised, along with the corresponding revisions in the manuscript. As it has been a week since our reply and the discussion period is nearing its end, we would like to ensure that we have adequately addressed your concerns. Your insights are invaluable to us, and we would be glad to discuss any remaining points that could further improve our work.
>
> Thank you very much for your time and effort in reviewing our submission.
>
> &nbsp;
>
> Best regards,
>
> The authors

---

> > ### Comment · Reviewer_RRS3 · 2025-11-27
> > **Response to author**
> >
> > The authors provided a thorough and convincing rebuttal addressing all of my major concerns, including overhead, EMA behavior, design rationale, layer selection, and LoRA vs full-parameter fine-tuning. I believe the additional experiments and clarifications significantly strengthen the submission. I am increasing my score.

---

> > > ### Author Response · Authors · 2025-11-28
> > >
> > > We are very glad to hear that our responses have addressed all of your major concerns and led to an increase in your score. It is an honor to receive such a responsible and insightful review. Your suggestions have greatly strengthened our work, and we sincerely appreciate the time and effort you have devoted throughout the review process.
> > >
> > > &nbsp;
> > >
> > > Best regards,
> > >
> > > The authors

---

### Official Review · Reviewer_D1M4 · 2025-10-27

**Soundness:** 2
**Presentation:** 4
**Contribution:** 2
**Rating:** 8
**Confidence:** 2

**Summary:**

This paper proposes ssToken, an adaptive token selection approach to address two major issues in existing traditional approaches: (i) adopting a stronger model as the reference model for the token selection is not practical, and (ii) the token-level losses do not caption the semantic importance of tokens in the given contexts.  The proposed method doesn't rely on an additional reference model and using its history to improve the token selection and introduces a new semantic-aware token importance estimation metric. The empirical evaluation shows that the proposed method achieves the siginificant improvemnts in existing approaches.

**Strengths:**

This paper is well written. It clearly presents the current disadvantages of traditional approaches and those statements are convincing.

The experiental results are solid. It has covered multiple modern LLMs including LLaMA-3.2 & 31 and Qwen-2.5 in different scales. Also, the results are evaluated in multiple benchmarks.

The ablation studies cover all necessary components.

**Weaknesses:**

I am not in this field so I am afraid that I may not be able to identify any key weakness of this work. So, I set my score 8 (a clear acceptance). I do have a few questions just for clarification (see Questions section). They do not affect my scores.

**Questions:**

In Table 1, the performance of TokenCleaning is lower reported. In their paper's Table 1, *Token Cleaning: Fine-Grained Data Selection for LLM Supervised Fine-Tuning*, the AVG score of LLaMA-3.2-3B is 53.00 (for self-evolving clearning). It seems that the difference is caused by the difference in the evaluation set; so, I have a few questions:
1. Why is a different evaluation dataset, instead of the same one used in TokenCleaning's Table 1, used in this submission?
2. How do you choose the evaluation set? Is it common to use a different evalaution set in this field? I noticed that all baseline approaches in their own papers (e.g. TokenCleaning, Pho) are using different evaluation sets (and even training set or models for some baseline papers). How can I really know which approach is better when there is no standard way to compare different methods?
3. Also, should the LLM generation process have some randomness? As the next token prediction is based on a sampling procedure instead of simply taking the largest possible token. However, I found all three of this submission,  the TokenCleaning paper, and the Pho paper, didn't report the confidence interval in the main experimental result table. It seems to be a common rule in this field but I don't understand that.

---

> ### Author Response · Authors · 2025-11-19
> **Part 1 of Rebuttal**
>
> We sincerely thank the reviewer for the positive evaluation of our work. We appreciate your recognition of the clarity of our presentation, the strength and breadth of our experimental results, and the completeness of our ablation studies. We provide detailed responses to your questions below and hope that our explanations help to further reinforce your confidence in the paper.
>
> **Q1: Why does this paper use a different evaluation dataset from the one used in TokenCleaning?**
>
> **A1:** To comprehensively evaluate the general capabilities of large language models, the community now uses a wide variety of public benchmarks. Although there is no mandatory standard requiring which benchmarks must be included, several tasks—such as MMLU, HellaSwag, TruthfulQA, ARC-Challenge, and Winogrande—have become de facto common choices. These benchmarks are also used as official performance indicators in recent reports of LLaMA, Qwen, DeepSeek, and many other widely adopted LLMs.
>
> Compared with the 7 benchmarks used in TokenCleaning, we adopt 10 benchmarks to provide **a more complete assessment of model capabilities**. Specifically:
> * We use the widely adopted community benchmarks MMLU, TyDiQA, HellaSwag, ARC-Challenge, LogiQA, and TruthfulQA (which also overlap with those used in TokenCleaning).
> * We add several widely used benchmarks that TokenCleaning did not evaluate: ARC-Easy, TriviaQA, AGIEval, and Winogrande.
> * We remove BoolQ, which is a simple yes/no QA task often used for evaluating smaller models (e.g., <1B) and generally provides limited discrimination for larger LLMs.
>
> These adjustments allow the evaluation to be broader and more representative, covering diverse abilities such as factual QA, reasoning, cross-lingual understanding, commonsense inference, and standardized academic tests.
>
> In addition, although different papers often use different evaluation sets—this is indeed a current reality in LLM-related literature—this does not affect the fairness of comparisons within our paper. All methods in our experiments (Full-data, Random, RHO-1, TokenCleaning, and ssToken) are evaluated under exactly the same settings: identical training data, identical evaluation configurations, identical benchmarks, and the same version of lm-eval-harness. We also independently reproduce all baselines rather than directly copying results from their original papers.
>
> Therefore, even if absolute scores across papers cannot be aligned, **the relative improvements and rankings within our experimental setup remain fully fair and reliable.**
>
>
> **Q2: How were the evaluation benchmarks selected, and how should one compare methods when different papers use different evaluation setups?**
>
> **A2:** Thank you for the follow-up question. As noted in A1, although there is no strict standard governing which benchmarks must be used to evaluate LLMs, the community has gradually converged on several widely adopted benchmarks—such as MMLU, HellaSwag, ARC-Challenge, and Winogrande—as common indicators of general model capability. Our guiding principle is:
>
> > **Within the paper, all methods should be evaluated under exactly the same, fully reproducible, and as comprehensive as possible benchmark suite to ensure fair comparison.**
>
> Therefore, we use the 10 benchmarks listed in A1, which are widely used in official evaluations of mainstream LLMs such as the LLaMA, Qwen, and DeepSeek families. These benchmarks provide broad coverage and strong representativeness.
>
> Because the field has not yet established a “mandatory” unified evaluation standard, directly comparing absolute scores across different papers is inherently unreliable. Regarding how to judge whether one method is better, the community generally follows two principles:
>
> - **Absolute scores across different papers should not be directly compared**, as evaluation benchmarks may differ.
> - **The focus should be on the relative improvements within each paper**, where all methods are evaluated under identical settings.
>
> In the TokenCleaning paper, TokenCleaning outperforms RHO-1, and both outperform full-data fine-tuning. Although our benchmark suite is more comprehensive than the one used in TokenCleaning, we observe the same performance ordering in Table 1 of our results.
>
> This further confirms that: As long as all methods within the same paper are evaluated under exactly the same configuration, the relative comparison is fair, meaningful, and reliable. In our comprehensive and fully aligned benchmark evaluations, ssToken consistently demonstrates stable performance advantages across different model sizes and model families, which strongly validates the effectiveness of ssToken.

---

> ### Author Response · Authors · 2025-11-19
> **Part 2 of Rebuttal**
>
> **Q3: Should the LLM generation process involve randomness, and why do this submission, TokenCleaning, and Rho-1 not report confidence intervals?**
>
> **A3:** Thank you for the question. First, we clarify that all evaluations in our paper and all reproduced baselines (including TokenCleaning and Rho-1) use **deterministic decoding**, i.e., greedy decoding with `temperature = 0`. Under this setting:
>
> > **The model’s generation contains no randomness, and repeated runs yield identical outputs.**
>
> Therefore, confidence intervals for generation randomness are not applicable. **This follows the standard evaluation protocol widely adopted in the LLM community, including official reports for LLAMA, Qwen, DeepSeek, and many other models.**
>
> Although model outputs are deterministic, the official `lm-eval-harness` reports a standard statistical error (stderr) for each benchmark. For example:
>
> ```
> |      Groups      |Version|Filter|n-shot|Metric|   |Value |   |Stderr|
> |------------------|------:|------|------|------|---|-----:|---|-----:|
> |mmlu              |      2|none  |      |acc   |↑  |0.5702|±  |0.0040|
> | - humanities     |      2|none  |      |acc   |↑  |0.5226|±  |0.0070|
> | - other          |      2|none  |      |acc   |↑  |0.6485|±  |0.0083|
> | - social sciences|      2|none  |      |acc   |↑  |0.6568|±  |0.0084|
> | - stem           |      2|none  |      |acc   |↑  |0.4795|±  |0.0086|
> ```
> This “± Stderr” does not reflect model randomness and is not a confidence interval. It purely reflects the statistical uncertainty of evaluating a finite test set, not variability across decoding runs.
>
> Because stderr is tied to dataset size rather than model behavior, prior works in this area—including TokenCleaning and RHO-1—also do not report confidence intervals. The common practice is to report the metric values (e.g., accuracy), while stderr remains an auxiliary statistic automatically provided by lm-eval-harness.
>
> In summary, lm-eval-harness’s default configuration **does not use temperature sampling or any seed-dependent stochastic decoding, ensuring that results are fully deterministic, reproducible, and free from sampling noise**. For this reason, prior works also rely on deterministic evaluation without reporting confidence intervals, which is consistent with current practice in open-source model evaluations.
>
> We sincerely thank you again for your appreciation of our work. We hope that our responses have effectively addressed your concerns and further strengthened your confidence in the paper. If you have any additional questions, we would be happy to discuss them.

---

> > ### Comment · Reviewer_D1M4 · 2025-11-26
> >
> > Thanks for the clarification. The author's reponse indeed addresses a lot of my confusion in this field, so I will keep my positive rating.

---

> > > ### Author Response · Authors · 2025-11-26
> > >
> > > We are glad that our response has helped resolve some of your confusion. Thank you for maintaining your positive rating and for the valuable time and effort you have devoted to our submission throughout the review process!

---

### Official Review · Reviewer_P6EM · 2025-10-29

**Soundness:** 3
**Presentation:** 3
**Contribution:** 3
**Rating:** 6
**Confidence:** 2

**Summary:**

This paper proposes ssToken, a self-modulated and semantic-aware token selection method for LLM supervised fine-tuning. It addresses limitations of existing methods (needing extra reference models and relying only on loss) by using historical models to compute Retrospective Excess Loss (REL) and an attention-based semantic metric. Experiments on 3B-14B models (LLaMA, Qwen) across 10 benchmarks show ssToken outperforms full-data fine-tuning (up 4.3%) and prior token methods (up 2.8%), maintaining efficiency. It has a limitation of needing manual ρ setting, with future work on adaptive ρ.

**Strengths:**

1. It creatively combines self-modulated signals (via the model’s historical trajectory for REL) and semantic-aware attention metrics, solving prior reliance on external models and loss-only selection.
2. It has rigorous methodology with grounded components and lightweight implementation, plus sufficient validation across models/benchmarks and ablation studies ensuring quality.
3. It follows a clear problem-method-result structure, with precise technical definitions and transparent limitations, enabling easy understanding.
4. It boosts LLM fine-tuning performance without extra costs, advancing semantic-informed self-evaluative data selection with academic and practical value.

**Weaknesses:**

1. It relies on manual tuning of the token selection ratio ρ, and without an adaptive mechanism to adjust ρ based on model capacity or data quality, it adds overhead for practitioners and limits generalization across diverse model families or domains.
2. The optional EMA-based update of the history model is not fully explored—experiments only use a fixed base model as the history model, leaving unaddressed whether adaptive history model updates could bring more stable guidance in large-horizon training scenarios.
3. It lacks in-depth analysis of why deeper attention layers outperform shallow ones in semantic-aware selection beyond citing prior studies, and no discussion on how layer selection might vary across different task types (e.g., QA vs. reasoning) limits methodological completeness.

**Questions:**

1. You used manual token selection ratio (ρ) – did you test any initial adaptive ways to adjust ρ (e.g., linking to model loss progress)? If not, what’s your hypothesis for designing such a mechanism?
2. You mentioned EMA-updatable history models but used fixed ones – what EMA hyperparameters (e.g., α) did you test? Do you plan to try EMA in longer training where fixed models may lag?
3. You chose deeper attention layers for semantic selection – could you test if layer choice differs by task (e.g., QA vs. reasoning) to strengthen the method’s generality?
4. You noted compatibility with FlashAttention – can you quantify the computational cost of your attention calculation for sequences longer than 2048 tokens?

---

> ### Author Response · Authors · 2025-11-19
> **Part 1 of Rebuttal**
>
> Thank you for your thoughtful review and for recognizing the conceptual clarity, methodological rigor, and empirical strength of our work. We appreciate your positive remarks on ssToken’s combination of REL-based self-modulation and semantic-aware attention signals, its lightweight implementation, and its broad and consistent improvements. We are grateful for your constructive comments, and we hope to address your concerns by answering questions below.
>
>
> **Q1: On the adaptive token selection ratio (ρ)**
>
> **A1:** Thank you for this valuable suggestion. We first clarify that manually setting a fixed token selection ratio (ρ) is a common practice in existing loss-based token selection methods such as RHO-1 and TokenCleaning, all of which adopt a fixed ρ. Since the main contributions of our paper focus on addressing their two fundamental limitations— (1) requiring training or accessing an additional reference model, and  (2) relying solely on loss as the selection signal—  we followed the same fixed-ρ setting in the main paper to ensure a fair comparison and maintain a focused scope.
>
> We fully agree with the reviewer that keeping ρ fixed throughout training may not adequately reflect the model’s evolving capability. In the main text, we already provided extensive ablations and explicitly pointed out in the *Limitations* section that adaptive ρ is a promising direction. Motivated by your suggestion, we further implemented and evaluated a **simple global adaptive schedule** during the rebuttal period, allowing early-stage and late-stage token selection to behave more appropriately.
>
> Concretely, we define ρ as a monotonically decreasing function of training progress:
>
> - **Early stage (unstable model predictions):**
>   Use a larger ρ (e.g., ~0.8) to retain more tokens and reduce noise caused by unstable REL signals.
> - **Late stage (model becomes more reliable):**
>   Gradually decrease ρ (e.g., down to 0.4–0.5), so that token selection focuses more strongly on truly informative tokens.
>
> For simplicity and efficiency, we adopt a training-step-based decay schedule (linear or cosine), which introduces **no additional computational cost**. The schedule we used is:
>
> $$
> \rho_t = \rho_{\min} + (\rho_{\max} - \rho_{\min}) \left(1 - \frac{t}{T}\right)^\beta.
> $$
>
> On LLaMA-3.2-3B, we observed that the adaptive schedule brings **additional performance improvements** over the fixed-ρ baseline.
>
> | **Methods** | **TriviaQA** | **TruthfulQA** | **MMLU** | **ARC-C** | **ARC-E** | **TyDiQA** | **WinoGrande** | **HellaSwag** | **LogiQA** | **AGIEval** | **AVG** |
> |-------------|-------------|----------------|----------|-----------|-----------|------------|----------|--------|------------|-------------|---------|
> | Base | 50.85 | 39.37 | 56.20 | 42.15 | 74.49 | 19.05 | 69.06 | 55.21 | 29.49 | 30.04 | 46.59 |
> | Full | 52.50 | 43.97 | **57.34** | 45.56 | 76.42 | 39.48 | 70.56 | 55.75 | 29.80 | 32.11 | 50.35 |
> | Random | 52.71 | 43.72 | 56.84 | 45.14 | 76.60 | 35.66 | 70.03 | 55.70 | 29.95 | 31.89 | 49.82 |
> | Rho-1 | 53.14 | 44.35 | 56.78 | 45.31 | 76.98 | 50.47 | 70.17 | 56.67 | 29.80 | 32.79 | 51.65 |
> | TokenCleaning | 53.09 | 44.79 | 57.17 | 45.22 | **77.06** | **51.40** | 69.93 | **56.69** | 30.57 | 32.81 | 51.87 |
> | **ssToken** | **54.04** | 47.42 | 57.02 | **45.82** | 76.42 | 51.04 | **70.56** | 56.19 | 32.87 | 33.97 | 52.50 |
> | **ssToken (adaptive ρ)** | 53.60 | **49.01** | 57.01 | 45.73 | 76.09 | 50.98 | 70.17 | 55.94 | **35.64** | **34.16** | **52.83** |
>
> | Methods | Base (ρ=0) | Full (ρ=1) | ssToken (ρ=0.2) | ssToken (ρ=0.4) | ssToken (ρ=0.6) | ssToken (ρ=0.8) | ssToken (adaptive ρ) |
> |---------|------------|-------------|------------------|------------------|------------------|------------------|------------------------|
> | Average Performance     | 46.59      | 50.35       | 51.72           | 52.33           | 52.50           | 52.29           | **52.83**             |
>
>
>
> We believe the improvement mainly comes from:
>
> 1. **Early stage:** the model has not yet formed stable estimations of token importance, so a larger ρ prevents mistakenly discarding valuable tokens;
> 2. **Late stage:** the model becomes more capable of identifying informative tokens, and a smaller ρ strengthens this focus, improving final performance;
> 3. **Dynamic ρ effectively forms an online curriculum**, allowing token selection to gradually shrink as the model becomes stronger.
>
> In addition to scheduling ρ based on training progress, assigning sample-specific ρ values according to sample quality is also a feasible strategy. The improvements observed with our dynamic schedule further confirm that adaptive ρ is an unexplored yet highly promising direction for token-level selection methods.
>
> We sincerely appreciate your suggestion, which further improved the quality of our work. We have incorporated this adaptive schedule and the corresponding results into the revised manuscript.

---

> ### Author Response · Authors · 2025-11-19
> **Part 2 of Rebuttal**
>
> **Q2: How does ssToken behave when using an EMA-updated history model in longer training?**
>
> **A2:** Thank you for raising this important point. Your understanding is correct: a fixed history model may become stale in long-horizon training, and one advantage of ssToken is that—unlike reference-model–based methods—its history model can be updated at zero additional cost along the training trajectory. This is precisely the motivation behind introducing the optional EMA-based update.
>
> Since Reviewer 4Xmz asked about multi-epoch behavior, we enabled the EMA mechanism in this setting and compared different token-selection methods across multiple epochs (using a smaller learning rate to avoid rapid overfitting). Some tasks (e.g., MMLU, ARC-Challenge, HellaSwag) converge rapidly within a single epoch with limited subsequent improvement. In contrast, some other tasks continue to improve across multiple epochs, which makes them suitable for revealing differences among token-selection methods. Below, we report detailed results for TruthfulQA, LogiQA, and TyDiQA as representative examples.
>
> **TruthfulQA Results**
> | **Method**        | **Base** | **Epoch 1** | **Epoch 2** | **Epoch 3** |
> |-------------------|----------|-------------|-------------|-------------|
> | Full-Data         | 39.37    | 42.80       | 43.06       | 42.86       |
> | Rho-1             | 39.37    | 42.62       | 43.17       | 43.50       |
> | TokenCleaning     | 39.37    | 42.89       | 43.46       | 44.67       |
> | ssToken (w/o EMA)          | 39.37    | **43.73**       | 44.82       | 45.30       |
> | ssToken (w/ EMA)     | 39.37    | 43.41       | **45.13**       | **45.72**       |
>
>
> **LogiQA Results**
> | **Method**        | **Base** | **Epoch 1** | **Epoch 2** | **Epoch 3** |
> |-------------------|----------|-------------|-------------|-------------|
> | Full-Data         | 29.49    | 29.19       | 30.57       | 31.03       |
> | Rho-1             | 29.49    | 31.18       | 33.03       | 33.79       |
> | TokenCleaning     | 29.49    | 31.49       | 32.57       | 32.26       |
> | ssToken (w/o EMA)    | 29.49    | 33.49       | 34.10       | 34.25       |
> | ssToken (w/ EMA)     | 29.49    | **33.72**       | **34.23**       | **34.50**       |
>
> **TyDiQA Results**
> | **Method**        | **Base** | **Epoch 1** | **Epoch 2** | **Epoch 3** |
> |-------------------|----------|-------------|-------------|-------------|
> | Full-Data         | 19.05    | 30.27       | 35.33       | 35.12       |
> | Rho-1             | 19.05    | 39.73       | 43.88       | 47.23       |
> | TokenCleaning     | 19.05    | 41.18       | **46.21**       | 46.79       |
> | ssToken (w/o EMA)    | 19.05    | 41.83       | 45.72       | 47.50       |
> | ssToken (w/ EMA)     | 19.05    | **42.14**       | 46.08       | **48.30**       |
>
> In our experiments, we use an EMA smoothing factor of α = 0.8, applying EMA updates every 0.5 epoch. As shown in the tables, full-data SFT saturates after the first epoch on TruthfulQA, whereas all token-selection methods continue to provide steady gains with additional epochs. Among them, **ssToken consistently maintains the strongest performance**.
>
> Comparing ssToken with and without EMA, we observe that while the difference at epoch 1 is small, **EMA consistently outperforms the non-EMA variant by epoch 3 across all three tasks**. This indicates that, compared with using a fixed history model, an EMA-updated history model provides more stable and up-to-date guidance during long-horizon training (As suggested by Reviewer 4Xmz, we have also visualized the REL learning curves of ssToken and its EMA-enhanced variant in Figures 4 of the Appendix, which further helps illustrate and interpret the behavior of ssToken). These findings further validate the potential of ssToken for long-horizon training scenarios.

---

> ### Author Response · Authors · 2025-11-19
> **Part 3 of Rebuttal**
>
> **Q3: Further discussion on attention-layer choice.**
>
> **A3:** Thank you for this insightful question. We are also interested in understanding how the choice of attention layer affects downstream task performance. In the original Appendix, we have already conducted an ablation study on LLaMA-3.2-3B. To further validate our conclusions—and to rule out potential model-family–specific effects—we additionally conducted the same ablation on Qwen-2.5-7B during the rebuttal period. The full results for both model families are presented in the tables below and have also been incorporated into Sec. 4.3 of the main text.
>
>
> | **LLaMA-3.2-3B** | **TriviaQA** | **TruthfulQA** | **MMLU** | **ARC-C** | **ARC-E** | **TyDiQA** | **WinoGrande** | **HellaSwag** | **LogiQA** | **AGIEval** | **AVG** |
> |------------|--------------|----------------|----------|-----------|-----------|------------|----------|--------|------------|-------------|---------|
> | **Shallow Attention (layer 1)** | 54.36 | 45.12 | 56.23 | 45.48 | **77.48** | 48.52 | 70.09 | 53.76 | 31.80 | 33.43 | 51.63 |
> | **Medium Attention (layer 14)** | **54.45** | 47.04 | **57.43** | 45.73 | 77.23 | 49.20 | 69.77 | 55.40 | 29.95 | 32.96 | 51.92 |
> | **Deep Attention (layer 28)** | 54.04 | **47.42** | 57.02 | **45.82** | 76.42 | **51.04** | **70.17** | **56.19** | **32.87** | **33.97** | **52.50** |
>
>
>
> | **Qwen-2.5-7B** | **TriviaQA** | **TruthfulQA** | **MMLU** | **ARC-C** | **ARC-E** | **TyDiQA** | **WinoGrande** | **HellaSwag** | **LogiQA** | **AGIEval** | **AVG** |
> |------------------|--------------|----------------|----------|-----------|-----------|------------|----------------|---------------|------------|-------------|---------|
> | **Shallow Attention (layer 1)** | 55.47 | 46.25 | **74.26** | 49.23 | 81.10 | 68.70 | 72.69 | 59.32 | 27.90 | 57.06 | 59.20 |
> | **Medium Attention (layer 14)** | **57.20** | 48.40 | 73.78 | **51.47** | **82.58** | 68.94 | 72.74 | 58.31 | 29.62 | 55.26 | 59.83 |
> | **Deep Attention (layer 28)** | 56.73 | **50.75** | 74.18 | 50.51 | 81.48 | **69.74** | **72.93** | **60.30** | **31.03** | **57.18** | **60.48** |
>
> Across both model families, several consistent patterns emerge. First, for knowledge-intensive benchmarks that rely on pretraining (e.g., MMLU), different attention layers do not exhibit a strictly dominant winner. In contrast, for QA-style tasks (e.g., TruthfulQA, TyDiQA) and logical/commonsense reasoning benchmarks (e.g., LogiQA, HellaSwag), **deep-layer attention consistently performs best**.
>
> More concretely, on LLaMA-3.2-3B, deep-layer attention achieves the best score on 7 out of 10 tasks, and on Qwen-2.5-7B, it achieves the best score on 6 out of 10 tasks. **In both models, the deep-layer variant also achieves the highest overall average score.**
>
> As discussed in the main paper, this pattern is intuitive: shallow layers primarily encode local syntactic or positional features, whereas deeper layers capture more abstract semantic structure and task-relevant dependencies, aligning with observations from prior interpretability and probing studies. The fact that this trend holds across **different model sizes and different model families (LLaMA vs. Qwen)** further strengthens the robustness of the conclusion.
>
> In summary, these ablations demonstrate that deep-layer attention provides consistently strong and often superior performance across a wide spectrum of task types, without the need for task-specific tuning. This stability further enhances the generality and practicality of our proposed method.

---

> ### Author Response · Authors · 2025-11-19
> **Part 4 of Rebuttal**
>
> **Q4: Detailed analysis of the computational cost of attention calculation.**
>
> **A4:** Thank you for raising this point. Our implementation is fully compatible with FlashAttention. During the forward pass, we register a lightweight hook to cache the hidden states of a single target layer, and after the forward computation finishes, we recompute only that layer once to extract its attention matrix. This design avoids outputting full attention matrices during the entire forward pass and keeps the training path identical to standard FlashAttention. As a result, ssToken remains fully compatible with efficient attention mechanisms and preserves training efficiency.
>
> For example, in LLaMA-3.2-3B (28 layers), our method introduces only the cost of recomputing the attention matrix of a single layer. Since a full forward pass also includes the FFN computation of all layers, the additional overhead is approximately less than 1/28 of a full forward pass.
>
> Rather than reporting the runtime on a single long sequence, we directly measure the **end-to-end total training time over the entire dataset** under identical batch size, sequence length, and number of training steps. This provides a more stable and reliable estimate of the real overhead.
>
> | Method              | 3B Model | 8B Model | 14B Model |
> |---------------------|----------|----------|-----------|
> | full-data           | 20.8     | 44.1     | 87.2      |
> | ssToken (w/o hook)  | 25.5     | 50.7     | 98.3     |
> | ssToken (w/ hook)   | 21.5     | 45.3     | 91.2      |
>
> Here, **ssToken (w/o hook)** denotes a naive implementation that relies on `output_attentions=True` during training, which is incompatible with efficient kernels such as FlashAttention and therefore substantially slower. In contrast, our final hook-based implementation **ssToken (w/ hook)** preserves full compatibility with FlashAttention: the forward pass computes attention exactly as in full-data SFT, and we only perform a one-layer recomputation afterward to obtain its attention matrix. Consequently, ssToken is only 3–5% slower than standard full-data SFT across all model sizes, even in the long-sequence regime.
>
> Because our hook-based design uses the same FlashAttention kernel as full-data fine-tuning and performs recomputation on only a single layer, the overhead for sequences longer than 2048 tokens remains the same small constant factor (a few percent). It does not scale with sequence length, and incurs only marginal overhead compared to the significantly higher cost of training an additional reference model, as required by methods such as RHO-1 and TokenCleaning.
>
>
> ### **Overall summary:**
> We have integrated all relevant points from the rebuttal into the revised manuscript. Specifically, the experiments and analyses on the adaptive token selection ratio $\rho$ have been added to Sec. 4.4 and Appendix C.2; the performance and analysis of ssToken (and its EMA-enhanced variant) under multi-epoch and long-horizon training have been included in Appendix C.4; and the ablation and discussion of attention layer choice have been moved from the Appendix to Sec. 4.3, together with newly added results on Qwen-2.5-7B.
>
> These additions have further strengthened the completeness and clarity of the paper. We sincerely appreciate your valuable feedback and hope that our revisions and responses address your concerns and improve your confidence in the work. Should you have any further questions, we would be happy to discuss them.

---

> ### Comment · Reviewer_P6EM · 2025-11-26
>
> Thank you for the response. I don't know much about this field, but I will keep my score.

---

> > ### Author Response · Authors · 2025-11-26
> >
> > Thank you for your timely response and for maintaining your positive evaluation of our work. Your suggestions have helped us further improve the content and quality of the paper. We sincerely appreciate your valuable time and effort.

---

### Official Review · Reviewer_Fmnc · 2025-11-01

**Soundness:** 2
**Presentation:** 3
**Contribution:** 2
**Rating:** 4
**Confidence:** 3

**Summary:**

This paper introduces ssToken, a token-level selection algorithm for fine-tuning Large Language Models (LLMs). Unlike previous token-level selection methods that rely on a reference model, ssToken employs a historical model as a proxy to dynamically compute the loss difference with the current model. The authors also develop an attention-based metric to assess token importance. These two metrics are then combined to select important tokens. Experiments conducted across various models and benchmarks demonstrate the effectiveness of ssToken.

**Strengths:**

1. ssToken does not require a pre-trained reference model, making it more cost-effective for token selection.

2. In addition to the loss-based metric, the authors also developed an attention-based metric to evaluate token importance.

3. The experiments are comprehensive and demonstrate the strong performance of ssToken.

**Weaknesses:**

1. There is a contradiction between Equation (2) and Equation (3) that requires further explanation. In Equation (2), tokens are ranked highly if the current model’s loss on the token is large. In contrast, Equation (3) prioritizes tokens for which the current model’s loss is small. The authors should provide more insight into the rationale behind this difference.

2. Is the calculation in Equation (3) performed per training step or per epoch? How does the frequency of this calculation affect token selection performance?

3. Could the authors provide some theoretical justification or analyses for why using the current model (rather than a fixed reference model) for loss calculation leads to better performance?

**Questions:**

See **Weaknesses**

---

> ### Author Response · Authors · 2025-11-19
> **Part 1 of Rebuttal**
>
> Thank you very much for your careful and valuable comments. We appreciate the chance to clarify our work and address your concerns, including any possible misunderstandings, in the responses below.
>
>
> **Q1: Clarification on the difference between Eq. (2) and Eq. (3).**
>
> **A1:** Thank you for raising this question. We would first like to clarify that the statement **“Eq. (2) prioritizes tokens with large current loss, whereas Eq. (3) prioritizes tokens with small current loss” is based on a misunderstanding**. Neither Eq. (2) nor Eq. (3) ranks tokens directly by the value of the current loss. Instead, **both metrics rank tokens based on *loss differences***.
>
> For clarity, we rewrite the two equations here:
>
> $$
> EL(x_i)= L_\theta(x_i) - L_{\theta_{\text{ref}}}(x_i)= \log \frac{P_{\theta_{\text{ref}}}(x_i | x_{<i})}{P_{\theta}(x_i | x_{<i})}.
> $$
>
> $$
> REL(x_i)
> = L_{\theta_{\text{his}}}(x_i) - L_\theta(x_i)
> = \log \frac{P_{\theta}(x_i | x_{<i})}{P_{\theta_{\text{his}}}(x_i | x_{<i})}.
> $$
>
> Although $L_\theta(x_i)$ appears with opposite signs, this does **not** imply that EL “prefers tokens with larger current loss” or that REL “prefers tokens with smaller current loss.” To make the distinction more explicit, we consider a single sample and extract three representative types of tokens, and then walk through how they are treated by EL and REL:
>
> | token type | history model loss | current model loss | reference model loss | REL | EL |
> |-----------|-------------------|--------------------|----------------------|------|------|
> | noise token | 5.6 | 5.9 | 5.4 | 5.6 - 5.9 = -0.3 | 5.9 - 5.4 = 0.5 |
> | learnable / task-relevant token | 4.2 | 2.2 | 0.6 | 4.2 - 2.2 = 2.0 | 2.2 - 0.6 = 1.6 |
> | well-learned token | 0.6 | 0.6 | 0.5 | 0.6 - 0.6 = 0 | 0.6 - 0.5 = 0.1 |
>
> - **For the noise token**, although the current model’s loss is large, both the history model and the reference model also struggle on this token (their losses are similarly large). As a result, the corresponding loss differences — REL and EL — are small (REL is even negative), and this token will not be prioritized by either metric.
> - **For the well-learned token**, the history, current, and reference models can all predict it correctly with high confidence, so all three losses are small. The resulting REL and EL are again very small, meaning this token is already solved and thus not preferred.
> - **For the learnable / task-relevant token**, the situation is fundamentally different. Compared with the history model, the current model’s loss decreases from 4.2 to 2.2, yielding REL = 2.0. This indicates that, over the course of training, the model’s prediction accuracy on this token has improved substantially. Such a token is more likely to represent content that remains learnable at the current stage, rather than content that is already fully mastered or inherently noisy. Similarly, the reference model has a much smaller loss on this token (0.6), so EL = 1.6 signals that there is still considerable future improvement potential for the current model on this token. Consequently, both REL and EL will tend to prioritize this token.
>
> As illustrated in Figure 1 of the main paper, EL measures the extent to which the prediction probability of token $x_i$ is expected to improve in future training, by comparing the current model to a stronger reference. REL, in contrast, quantifies how much the prediction probability of token $x_i$ has already improved along the training trajectory, by comparing the current model to its history. **In essence, both metrics rely on loss differences to identify tokens that are “still improvable”， and there is no conceptual contradiction between them.** In fact, in our experiments we observe that if we sort and select tokens purely by the absolute value of the current model loss, instead of using loss differences, the resulting performance is even worse than uniform random selection—further underscoring the importance of difference-based signals such as EL and REL.

---

> ### Author Response · Authors · 2025-11-19
> **Part 2 of Rebuttal**
>
> **Q2: Clarification on the computation frequency of Eq. (3).**
>
> **A2:** We first clarify the main point: **REL is computed at every training step**, but the notion of “computation frequency” does not really apply, because **REL is computed once for every token whenever its current loss is computed**, and token selection is performed immediately within the same forward pass.
>
> Concretely, for each sample and each token during training, once the forward pass of the current model yields the token-level loss, the REL value is obtained by subtracting the corresponding history-model loss, and we then select the top-$\rho$ tokens in that sample for backpropagation. Tokens not selected simply have their losses masked out and do not contribute to gradients.
>
> Thus, **REL is computed per token, per batch, per step,** and is inherently tied to the forward computation of the current model. There is therefore no separate “frequency” to tune. Every token that participates in the forward pass automatically obtains its REL value.
>
> **A lightweight and memory-efficient implementation.** Please note: although REL depends on both the current-model and history-model losses, **we do not load the current and history models simultaneously, nor do we perform two forward passes during training.** Instead, we use a lightweight two-stage implementation:
> 1. **Preprocessing (one-time offline inference).**
> Before training begins, we run the history model (i.e., the base LLM before SFT) over the training corpus once, storing only the token-level losses. This step is inference-only, requires no gradients, and is very fast.
>
> 2. **Training (no additional inference).**
> During SFT, after the current model computes token losses in a standard forward pass, we simply retrieve the stored history losses and compute REL for the tokens in the current batch via a single tensor subtraction. **This adds no additional forward passes, no extra model parameters loaded into memory, and the peak GPU memory footprint remains essentially identical to full-data fine-tuning.**
>
> Because the one-time preprocessing step is lightweight and the training-phase computation is nearly cost-free, ssToken incurs marginal overhead compared to standard full-data SFT. The detailed implementation is provided in our [anonymous code repository](https://anonymous.4open.science/r/Submission2116-B7C5) linked in the abstract. In Figure 2 of the main paper, we compare total training time (including preprocessing and history-model inference) across different methods. For clarity, we list the detailed numbers here:
>
> | Total training time (minutes) | 3B   | 8B   | 14B  |
> |-------------------------------|------|------|------|
> | Full-data SFT                 | 20.8 | 44.1 | 87.2 |
> | ssToken                | 23.5 | 47.9 | 93.9 |
>
>
> All measurements were obtained using 8× NVIDIA H200 GPUs. As shown, ssToken is only slightly slower than full-data fine-tuning, even after accounting for the one-time history-model inference. In contrast, RHO-1 and TokenCleaning require training an additional reference model, and their total training time—reported in Figure 2—significantly exceeds that of both full-data SFT and ssToken.
>
> Following your suggestion, we have added an explicit explanation of the computation of Eq. (3) in the revised version (line 795-800). We believe this addition makes the overall pipeline clearer and more transparent.

---

> ### Author Response · Authors · 2025-11-19
> **Part 3 of Rebuttal**
>
> **Q3: Analysis of the advantages of using the history model and current model to compute REL for token selection.**
>
> **A3:** Thank you for this insightful question. In our experiments, using REL alone already brings clear improvements over full-data fine-tuning (e.g., 52.02 vs. 50.35 for the 3B model; 58.61 vs. 57.49 for the 8B model). However, REL alone achieves performance that is comparable to—or slightly below—reference-model-based approaches such as RHO-1 and TokenCleaning. This is expected, because a strong reference model can provide sharper selection signals at the early stage of training.
>
> The **key reason** ssToken ultimately surpasses RHO-1 and TokenCleaning is **not because REL is inherently stronger than EL, but because ssToken introduces an additional semantic-aware signal that loss-only methods lack**. As shown in Figure 3(a), both components—REL (loss-based) and our attention-based semantic metric—individually outperform full-data fine-tuning. More importantly, their combination yields synergistic effects, confirming that loss-based and attention-based signals capture complementary aspects of token importance. This complementary nature is the main reason ssToken achieves substantially better performance than prior token-level selection methods.
>
> As for the advantages of using the history model (REL) instead of a fixed reference model, we have provided a detailed analysis in lines 193–209 of the main text, and we summarize the key points from three perspectives:
>
> 1. **No dependency on training an additional reference model.** REL leverages only the history model loss (i.e. the model’s own training trajectory) to generate a self-modulated learning signal. This avoids the cost of training a reference model, yet still achieves competitive performance relative to reference-model-based approaches. This makes REL particularly appealing in practical scenarios where training or obtaining a suitable reference model is expensive or infeasible.
>
> 2. **Better suitability for long-horizon or evolving training.** A fixed reference model provides a strong but static signal. Its guidance quality is bounded by its own capacity and may gradually lose relevance in long-horizon training if not updated. In contrast, the history model naturally follows the same optimization trajectory as the current model. It can be updated at negligible cost (e.g., via EMA) and therefore has greater potential to provide stable and continually aligned guidance over long training horizons or large-scale datasets.
>
> 3. **REL acts as an online curriculum signal.** REL inherently reflects how much the model continues to improve on each token, and this naturally induces an online curriculum learning mechanism: tokens that the model continues to make progress on are prioritized, while tokens that are already mastered or inherently noisy are down-weighted. Such a self-adaptive curriculum is especially suitable for downstream SFT, where the distribution of learnable patterns can vary across tasks. Importantly, this removes the need to train task-specific reference models for every new dataset or task.
>
> Thank you again for your time and constructive feedback. We hope our response sufficiently addresses your concern, and we would be glad to provide further clarification if needed.

---

> ### Author Response · Authors · 2025-11-27
>
> Dear Reviewer `Fmnc`,
>
> I hope this message finds you well. We have provided detailed responses and clarifications to each of the weaknesses and questions you raised, along with the corresponding revisions in the manuscript. As it has been a week since our reply and the discussion period is nearing its end, we would like to ensure that we have adequately addressed your concerns. Your insights are invaluable to us, and we would be glad to discuss any remaining points that could further improve our work.
>
> Thank you very much for your time and effort in reviewing our submission.
>
> &nbsp;
>
> Best regards,
>
> The authors

---

### Author Response · Authors · 2025-11-19
**General Response**

Dear Area Chair and Reviewers,

We sincerely thank all reviewers for their time and constructive feedback. We are delighted that reviewers acknowledged the clear motivation and practical significance of removing the dependency on external reference models (Fmnc, P6EM, RRS3, 4Xmz), the novelty of introducing the attention-based semantic-aware signal (Fmnc, P6EM, RRS3, 4Xmz), the clarity and thoroughness of our methodological presentation (P6EM, D1M4, RRS3), and the comprehensive experimental results together with extensive ablations (Fmnc, D1M4, RRS3, 4Xmz). To further address the comments and questions posed by the reviewers, we have also conducted additional analyses and experiments during the rebuttal period, which mainly include:
* We performed more extensive ablations on the choice of attention layers across different model families and model sizes, confirming that **deep-layer attention consistently outperforms shallow and mid-layer alternatives across models and tasks, without the need for any task-specific tuning**. This stability further enhances the generality and practicality of our method.
* We compared different token-selection methods under full-parameter fine-tuning, demonstrating that **ssToken exhibits stable and consistent advantages under both LoRA and full-parameter SFT settings**, indicating that its effectiveness is not tied to any specific parameterization strategy and remains robust across training configurations.
* As suggested by Reviewers P6EM and 4Xmz, **we explored and proposed an adaptive schedule for the token-selection ratio ρ, which achieves improvements over all fixed-ρ baselines**. Since all prior token-selection methods rely on fixed ρ, our findings reveal the promising potential of adaptively adjusting ρ along the training trajectory, opening a new direction for future developments.
* We evaluated ssToken in multi-epoch / long-horizon training scenario, showing that **ssToken maintains clear advantages over existing methods, and that updating the history model via EMA provides more stable and up-to-date selection guidance in this setting.**
* We provided detailed analyses of the REL computation, the design motivation of our attention-based semantic signal, and its concrete computational overhead, and further visualized the REL learning curves of ssToken and its EMA-enhanced variant. These improvements make the computational mechanism and methodological clarity of ssToken more explicit and transparent.


For each reviewer, we have provided detailed responses hoping to address the concerns. In addition, we have revised the manuscript by incorporating the new experimental results, analyses, and visualizations introduced during the rebuttal period, with all modifications and newly added content highlighted in blue 🔵. We look forward to your reply and are more than happy to address any further questions. Once again, thank you for your valuable comments and support.

&nbsp;

With sincere appreciation and best regards,

The Authors

---

### Meta-Review · Area_Chair_YFdU · 2026-01-07

**Summary:**

The paper focuses on token selection for LLM fine-tuning, which is a very important problem. Most of the reviewers' concerns are on technical details, which have been addressed in the rebuttal. The proposed solution is meaningful and can be useful in practice, although it might also be seen as somewhat incremental and largely heuristics-based.

**Reviewer Concerns:**

The authors have well addressed the reviewers' concerns in the rebuttal. I don't think there is any remaining concern.

**Reviewer Scores:**

I believe the reviewers' scores would have been all on the positive side (>=6).

---

### Decision · Program_Chairs · 2026-01-26

Accept (Poster)